# An artificially-intelligent cornea with tactile sensation enables sensory expansion and interaction

Shangda Qu[1,2,3], Lin Sun[1,2,3], Song Zhang[1,2], Jiaqi Liu[1,2], Yue Li[1,2], Junchi Liu[1,2] & Wentao Xu ®[1,2] ✉

We demonstrate an artificially-intelligent cornea that can assume the functions of the native human cornea such as protection, tactile perception, and light refraction, and possesses sensory expansion and interactive functions. These functions are realized by an artificial corneal reflex arc that is constructed to implement mechanical and light information coding, information processing, and the regulation of transmitted light. Digitally-aligned, long and continuous zinc tin oxide (ZTO) semiconductor fabric patterns were fabricated as the active channels of the artificial synapse, which are non-toxic, heavy-metal-free, low-cost, and ensure superior comprehensive optical properties (transmittance >99.89%, haze <0.36%). Precisely-tuned crystal-phase structures of the ZTO fibers enabled reconfigurable synaptic plasticity, which is applicable to encrypted communication and associative learning. This work suggests new strategies for the tuning of synaptic plasticity and the design of visual neuroprosthetics, and has important implications for the development of neuromorphic electronics and for visual restoration.

Located at the front of the eye, a cornea is a transparent structure that provides focusing power and shields the iris and lens from foreign substances[1]. It is the most densely-innervated part of the body, and is therefore sensitive to foreign contaminants; a touch of the cornea causes an involuntary reflex to close the eyelid (the corneal reflex)[2]. However, corneal diseases can result in blindness, and more than 10 million individuals worldwide suffer from bilateral corneal blindness[3–5]. Corneal transplantation of a donor allograft (i.e., keratoplasty) is the most common medical solution to this disease[6–8], but due to scarcity of donated corneas, it can be used on only 1 in 70 patients[9–11], and 12.7 million patients are waiting for the process[12,13].

As a solution to this scarcity, artificial corneal substitutes that comply with the requirements for optical and transparency have been developed[14]. The Boston keratoprosthesis (Kpro) is the most-commonly implanted artificial corneal substitute worldwide; it is a collar-button-shaped device that is mainly made of polymethyl methacrylate (PMMA)[15–17]. Aurolab Kpro shares the design with Boston Kpro and is considered an alternative to the latter when affordability is a limiting factor[18]. MICOF Kpro contains a central ring with a threaded PMMA optic that is supported by a titanium frame, and the surgical invasiveness is mild[19–21]. As a unique approach to the artificial cornea, the osteo-odonto-keratoprosthesis is a biological Kpro that uses the lamina derived from an autologous mono-radicular tooth as a frame for PMMA optical cylinder[22,23]. All of these types of artificial cornea have been applied clinically to improve vision in cases of severe corneal opacification, and the effects of visual restoration, long-term safety, practicality, and cost are acceptable to the patients.

Although existing artificial corneas can assume partial functions of the native human cornea, such as protection and light refraction, they do not reconstruct the tactile sensation and therefore do not

[1]Institute of Photoelectronic Thin Film Devices and Technology, Key Laboratory of Photoelectronic Thin Film Devices and Technology of Tianjin, College of Electronic Information and Optical Engineering, Engineering Research Center of Thin Film Photoelectronic Technology of Ministry of Education, Smart Sensing Interdisciplinary Science Center, Nankai University, Tianjin 300350, China. [2]Shenzhen Research Institute of Nankai University, Shenzhen 518000, China. [3]These authors contributed equally: Shangda Qu, Lin Sun. ✉e-mail: wentao@nankai.edu.cn

realize the corneal reflex. Thus, trying to develop a 'smarter' artificial cornea with tactile sensation and even possessing sensory expansion and interactive functions is of great significance for restoring vision in corneal blindness. Moreover, these functions must be realized by 'invisible' built-in electronics[24–28], to achieve the high transparency and low haze of a natural cornea.

This paper describes an artificially-intelligent cornea that has the functions of protection, tactile perception and light refraction like the native human cornea, and also has sensory expansion (widening an existing sensory experience) and interactive functions. The touch sensitivity of the cornea was reconstructed by a reflex arc composed of sensor-oscillation circuits, zinc tin oxide (ZTO) artificial synapses (ASs) and electrochromic devices, which implement mechanical and light information coding, information processing and the regulation of transmitted light. Heavy-metal-free ZTO fabric patterns were fabricated with long, continuous morphology and digital alignment, and used as active channels due to their non-toxicity, low-cost, high transmittance (>99.89%), and low haze (<0.36%) to ensure the optical properties of the artificial synapse. The crystal phase of the ZTO fibers was highly tuned, and therefore ensured tunable synaptic plasticity, and incidental applications to encrypted communication and associative learning. This work presents a new resource that may be used in development of powerful neuromorphic electronics and visual neuroprosthetics.

## Results

### Overview of an artificial corneal reflex arc

The corneal reflex is an involuntary protective eyelid closure in response to mechanical stimulation or light flashes[29–31]. The neural pathway of the corneal reflex is a loop between the trigeminal sensory nerves and the facial motor nerve innervation of the orbicularis oculi muscles[32] (Fig. 1). To construct an artificial corneal reflex arc at the level of a neural pathway, the system was fabricated of three core components: sensor-oscillation circuits as the receptors that transform external stimuli to impulse spikes, ZTO ASs as the processing core that transfers and integrates information, and electrochromic devices as

the actuator that respond to the postsynaptic current (Fig. 1). Trigeminal sensory nerves transfer signals through synaptic connections (Fig. 2a); an artificial corneal reflex arc must mimic this pathway.

### Digitally-aligned ZTO fibers with tunable crystal structure

To mimic biological synapses, ion-gel-gated synaptic transistors were developed. They were composed of source and drain electrodes, digitally-aligned ZTO fibers channel and an electrolyte that uses poly(vinylidene fluoride-co-hexafluoropropylene) (PVDF-HFP) (Fig. 2b). ZTO ASs that used different Zn:Sn molar ratios (3:7, 1:1 and 7:3) were evaluated. Artificial excitatory postsynaptic currents (aEPSCs) of the ZTO-3:7 AS were measured by applying a single presynaptic spike (2 V, 50 ms) at drain voltage $V_d = 0.5$ V (Fig. 2c). The rise time of the aEPSC is related to the settings in the test of synaptic devices, and is generally positively correlated with the duration of the presynaptic spike. Upon the arrival of positive presynaptic spikes, cations in the PVDF-HFP electrolyte accumulated at the interface between ion gel and ZTO fibers, or become embedded in the defects and oxygen vacancies of ZTO fibers; these changes resulted in formation of an electric double layer. This ion/electron electrostatic coupling effect induced accumulation of carriers in the ZTO fibers and thereby caused an increase in postsynaptic current, as measured from the drain electrode[33]. After the presynaptic spikes stopped, the cations diffused back to a random distribution in the PVDF-HFP electrolyte, so the aEPSC gradually decayed to a low level. Notably, 10 s after removal of the spike, the aEPSC of the ZTO-3:7 AS triggered by a single spike decayed to 118% of the quiescent current, rather than to the initial value. This phenomenon can be attributed to trapping of ions in defects and oxygen vacancies within the ZTO fibers. These trapped ions return slowly to their initial distribution, so an aEPSC is retained even after the presynaptic spike ends[34].

ZTO fibers were fabricated using an electrohydrodynamic nanowire printer that has a digitally-controlled alignment mechanism. Structural and chemical properties of ZTO fibers with different Zn:Sn molar ratios and their corresponding synaptic performance were measured (Supplementary Table 1). Scanning electron microscopy

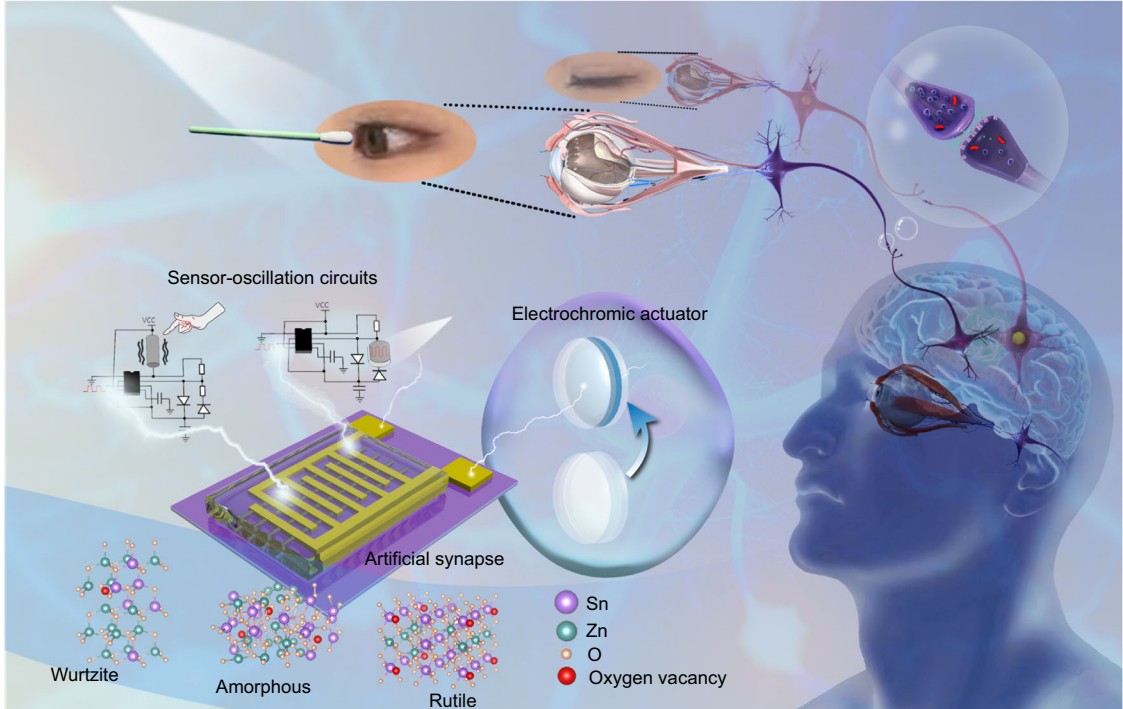

**Fig. 1 | Overview of an artificial corneal reflex arc.** Schematic of the corneal reflex and the corresponding artificial corneal reflex arc composed of sensor-oscillation circuits, ZTO ASs, and electrochromic devices.

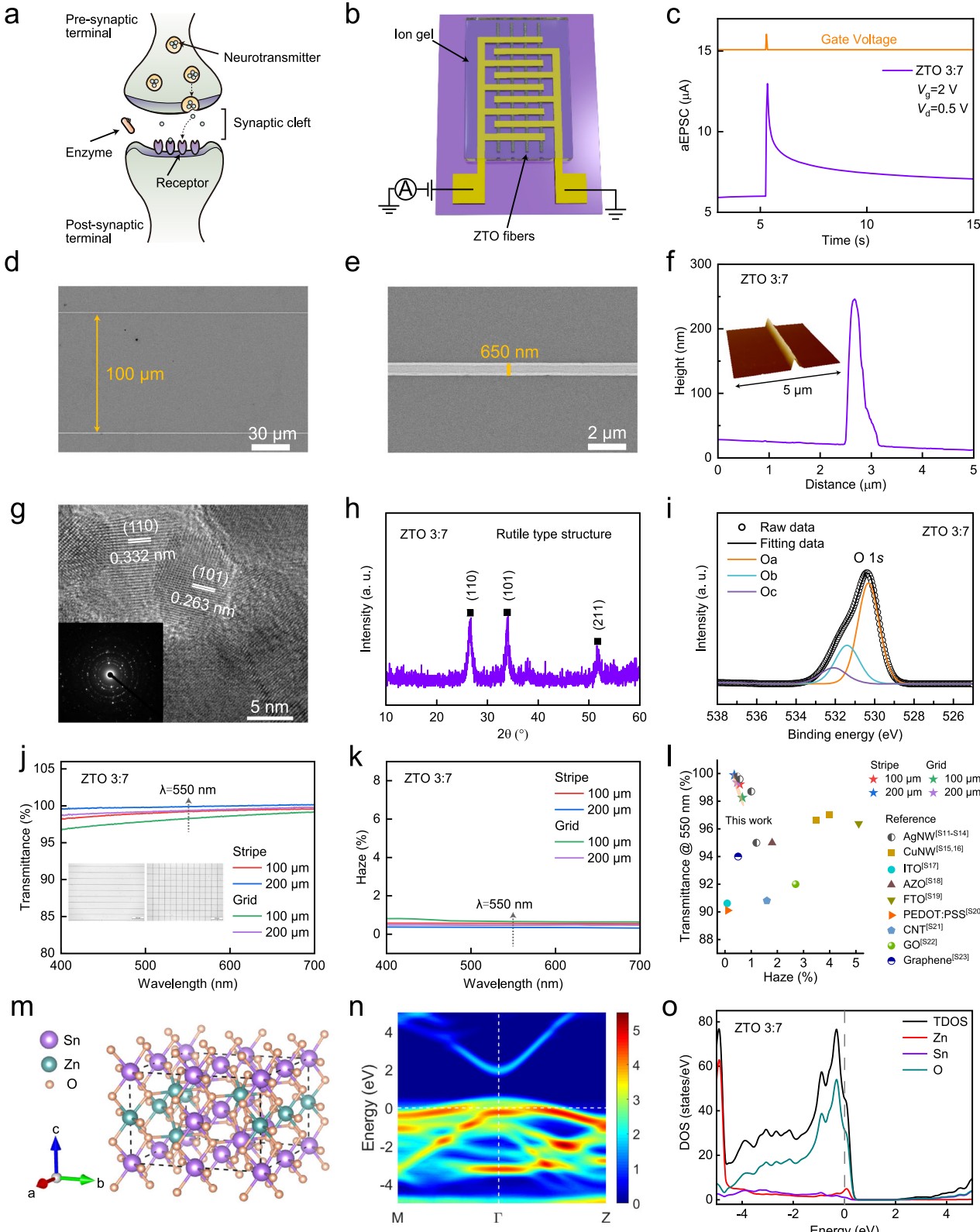

**Fig. 2 | Digitally-aligned ZTO fibers with tunable crystal structure. a** Schematic of a biological synapse. **b** AS that uses ZTO fibers. **c** aEPSC of ZTO-3:7 AS triggered by a single spike. **d, e** SEM images of ZTO-3:7 fibers array (**d**) and a single fiber (**e**). **f** Cross-sectional analysis of a single ZTO-3:7 fiber. Inset: AFM image of a single ZTO-3:7 fiber. **g** HRTEM image of ZTO-3:7 fibers. Inset: SAED pattern of ZTO-3:7 fibers. **h** XRD pattern of ZTO-3:7 fibers. **i** O 1$s$ XPS spectra of ZTO-3:7 fibers. **j, k** Visible-range transmittance (**j**) and haze (**k**) of ZTO-3:7 fibers with different patterns and pitch sizes. Inset: OM images of the ZTO fibers. **l** Transmittance (at wavelength 550 nm) versus haze for ZTO-3:7 fibers and other materials. **m** Optimized structure of ZTO-3:7 fibers. **n** Energy band structure of ZTO-3:7 fibers. **o** DOS of ZTO-3:7 fibers. The Fermi level is set to zero.

(SEM) was conducted to characterize the surface morphology of ZTO-3:7 fibers. The spacing between digitally-aligned, long and continuous ZTO-3:7 fibers was ~100 μm (Fig. 2d), and their typical diameter was ~650 nm (Fig. 2e). A ZTO AS used ~19 ZTO fibers as channels, and the total contact area between ZTO fibers and ion gel was ~6792.5 μm² (Supplementary Fig. 1). The aEPSC increased as the number of ZTO fibers increased (Supplementary Fig. 2). Atomic force microscopy (AFM) image (inset of Fig. 2f) and cross-sectional analysis (Fig. 2f) of a single ZTO-3:7 fiber demonstrated its 3D shape with a height of ~200 nm. High-resolution transmission electron microscopy (HRTEM) of the ZTO-3:7 fibers (Fig. 2g) shows well-resolved lattice fringes with spacings of ~0.332 nm and ~0.263 nm, which are attributed to (110) and (101) planes, respectively. The selected area electron diffraction (SAED) pattern of ZTO-3:7 fibers confirmed their polycrystalline structure (inset of Fig. 2g). The X-ray diffraction (XRD) pattern (Fig. 2h) has three sharp peaks at $2\theta \approx 26.62°$, $33.93°$ and $51.75°$; this result also demonstrates that the ZTO-3:7 fibers have a polycrystalline structure. Pure crystalline $SnO_2$, which has a rutile structure (JCPDS No. 41–1445), has XRD peaks at $26.61°$, $33.89°$ and $51.78°$, which correspond to (110), (101) and (211) planes, respectively. The similarity of these angles to those of the ZTO-3:7 fibers indicates that they have a rutile structure; the small differences can be attributed to doping of Zn atoms into $SnO_2$. X-ray photoelectron spectroscopy (XPS) was used to determine the chemical states of ZTO fibers. The O 1$s$ peak of ZTO-3:7 fibers was divided into three peaks, approximately centered at 530 eV ($O_a$), 531 eV ($O_b$), and 532 eV ($O_c$) (Fig. 2i). The $O_a$ peak was related to metal-oxide bonds, the $O_b$ peak was assigned to oxygen vacancies and is beneficial to mimic behaviors of biological synapses, and the $O_c$ peak was related to oxygen in impurities such as hydroxyl groups. For study of the optical properties of ZTO fibers, they were deposited in stripe and grid patterns with pitch sizes of 100 and 200 μm by adjusting the printing parameters. The pitch sizes can meet the requirements for constructing an artificial nervous system[35]. Optical microscopy (OM) images distinguished the patterns (inset of Fig. 2j). At the wavelength $\lambda = 550$ nm, the stripe ZTO-3:7 fibers (200 μm) had the highest transmittance (99.89%) and the lowest haze (0.36%) (Fig. 2j, k). Besides the virtues of heavy-metal-free, non-toxicity, and low-cost, the comprehensive optical properties of ZTO fibers are superior to other materials such as metallic nanowires, transparent oxide films, poly(3,4-ethylenedioxythiophene):poly(styrene sulfonate) (PEDOT:PSS) films, carbon nanotubes and graphene (Fig. 2l, Supplementary Table 2); this result demonstrates the applicability of ZTO fibers in transparent electronics. The transparency of the entire synaptic device prepared on a glass substrate was ~78.91% at $\lambda = 550$ nm (Supplementary Fig. 3).

The structural, chemical and optical properties of ZTO fibers with Zn:Sn molar ratios of 7:3 and 1:1 were also measured. SEM images (Supplementary Fig. 4), AFM images, and cross-sectional analysis (Supplementary Fig. 5) of ZTO-7:3 and ZTO-1:1 fibers showed surface morphology, 3D shape, and height that were consistent with those of ZTO-3:7 fibers. However, the XRD patterns of ZTO-7:3 and ZTO-1:1 fibers are markedly different from ZTO-3:7 fibers. For ZTO-7:3 fibers, the XRD pattern shows four diffraction peaks at $2\theta \approx 31.73°$, $34.44°$, $36.27°$ and $56.62°$ (Supplementary Fig. 6a), which correspond to the (100), (002), (101) and (110) reflections of wurtzite ZnO (JCPDS No. 36–1451). In contrast, for ZTO-1:1 fibers (Supplementary Fig. 6b), the XRD pattern shows no obvious sharp peaks; i.e., these fibers have an amorphous structure. The polycrystalline structure of ZTO-7:3 fibers and the amorphous structure of ZTO-1:1 fibers was confirmed by HRTEM images and SAED patterns (Supplementary Fig. 7). The structure formation of ZTO-1:1 fibers may occur because the distinct crystallographic arrangements of rutile and wurtzite crystal structures lead to an increase in the energy barriers to nucleation and diffusion[36]. O 1$s$ spectra of ZTO-7:3 and ZTO-1:1 fibers were obtained using XPS (Supplementary Fig. 8). In amorphous ZTO, the oxygen associated with under-coordinated metal ions can be referred to the $O_b$ peak, which is related to oxygen vacancies. The area ratio of $O_b$ peaks increased monotonically as Zn:Sn molar ratio decreased: ~18% in ZTO-7:3, ~30% in ZTO-1:1, and ~43% in ZTO-3:7. This trend occurs because the Sn-O bond is weaker than the Zn-O bond, so decreasing Zn:Sn molar ratio induces increase in the numbers of oxygen vacancies and under-coordinated metal ions[37,38]. The binding energy of Sn $3d_{5/2}$ peaks was centered at 486.5-486.6 eV; this result indicates that the Sn ions in ZTO fibers are mainly present as $Sn^{4+}$ (Supplementary Fig. 9). The optical transmittance of different ZTO fibers were very similar (Supplementary Fig. 10). This result indicates that the differences in crystal structure had little effect on this trait.

Double-sweep transfer characteristics (from −2 to 4 V) of ZTO-7:3, ZTO-1:1 and ZTO-3:7 ASs were measured at $V_d = 0.5$ V (Supplementary Fig. 11a). All three devices show anticlockwise hysteresis of drain current with a memory window; this result indicates that the carrier density in ZTO channels can be effectively adjusted by the mobile ions. The aEPSCs of ZTO-7:3 and ZTO-1:1 ASs were measured after application of a single presynaptic spike (2 V, 50 ms) at $V_d = 0.5$ V (Supplementary Fig. 11b, c). The aEPSC peak value increased monotonically as the Zn:Sn molar ratio decreased from 7:3 to 3:7. Furthermore, the aEPSC decayed to its initial value faster in ZTO-7:3 and ZTO-1:1 ASs than in the ZTO-3:7 AS. The difference in electrical characteristics of the three devices occurs because the decreasing Zn:Sn molar ratio causes increase in numbers of oxygen vacancies and partially ionized oxygen vacancies. The increase in the number of oxygen vacancies increases the number of trap sites for mobile ions, so aEPSC is retained; the partially ionized-oxygen vacancies act as shallow donors, and increase carrier density of the ZTO channel to increase the aEPSC peak. The field-effect carrier mobility $\mu_n$ [cm²/(V·s)] was calculated as 8.9 for ZTO-7:3, 6.3 for ZTO-1:1, and 12.8 for ZTO-3:7 ASs (Supplementary Note 1, Supplementary Fig. 12).

To understand the structural properties of ZTO fibers with different Zn:Sn molar ratios, first-principles calculations were conducted using density functional theory (DFT) in the Vienna Ab-initio Simulation Package[39,40]. Three $2 \times 2 \times 2$ supercells that corresponded to ZTO fibers with the three Zn:Sn molar ratios were constructed. The initial structures for the ZTO-7:3 and ZTO-3:7 fibers were selected as wurtzite and rutile, respectively. After geometry optimization, the structures with the ZTO-7:3 and ZTO-3:7 fibers were slightly disordered (Supplementary Fig. 13a and Fig. 2m). The structure of ZTO-1:1 fibers was stimulated using ab initio molecular dynamics (AIMD), with rutile type as the initial structure. The annealing temperature in the simulation was consistent with the annealing temperature used during the fabrication of ZTO fibers. In the simulation, after annealing, the structure of ZTO-1:1 fibers was more disordered than those of ZTO-7:3 and ZTO-3:7 (Supplementary Fig. 13b). The band structures were calculated for the ZTO-7:3 (Supplementary Fig. 14a), ZTO-1:1 (Supplementary Fig. 14b) and ZTO-3:7 (Fig. 2n) fibers. The band dispersions were indistinguishable, especially for the amorphous structure of ZTO-1:1 fibers. The density of states (DOS) for the ZTO-7:3 (Supplementary Fig. 15a), ZTO-1:1 (Supplementary Fig. 15b), and ZTO-3:7 (Fig. 2o) fibers shows that the band gaps of three samples were 0.75, 0.53 and 1.58 eV, respectively, which are similar to other calculations[41,42]. The simulation values are smaller than the corresponding experimental values of 3.62, 3.54, and 3.66 eV (Supplementary Note 2, Supplementary Fig. 16); this difference can be attributed to the limitation of DFT[41,43]. However, the variation trend of band gaps obtained from experiments is consistent with that from the simulation (Supplementary Table 3). The band gaps increased as Zn:Sn molar ratio decreased. The band gap of ZTO-1:1 fibers was the smallest, because amorphization of the crystal structure induces defects in the system[44]. In addition, DOS at the Fermi level (set to zero) also increased as Zn:Sn molar ratio decreased; this trend indicates an increase in the conductivity of ZTO fibers. These phenomena are consistent with the increase in aEPSC of ZTO ASs as Zn:Sn molar ratio decreases.

## ZTO-fibers artificial synapses with tunable synaptic plasticity

Paired-pulse facilitation (PPF) is a typical synaptic feature in a biological system. In PPF, the amplitude of aEPSC is higher after the second of two consecutive presynaptic spikes than after the first[45,46] (Fig. 3a). PPF of three ZTO ASs was quantified by applying a pair of spikes (2 V, 50 ms) separated by different time intervals $\Delta t$ (Fig. 3b). At $\Delta t = 100$ ms, the amplitude of the second aEPSC peak (A2) was obviously bigger than the amplitude of the first aEPSC peak (A1) (Fig. 3c, inset of Supplementary Fig. 11b, c). Experiments performed regarding synaptic plasticity were repeated more than three times. Corresponding statistics information was shown as mean values and standard deviations. PPF index [%] $I_{PPF} = 100 \cdot A2/A1$ of ZTO-7:3, ZTO-1:1, and ZTO-3:7 ASs all decreased gradually as $\Delta t$ increased; this trend occurs because of the increase in time available for cations to diffuse back to their initial distributions (Fig. 3d). The mean maximum $I_{PPF}$ (at $\Delta t = 50$ ms) were 133.0% for ZTO-7:3, 127.7% for ZTO-1:1, and 132.5% for ZTO-3:7ASs.

Spike-number-dependent plasticity (SNDP) of three ZTO ASs was obtained by applying consecutive spikes (2 V, 50 ms) with different spike numbers ($n_S = 5, 10, 20, 30, 50$) at $V_d = 0.5$ V (Fig. 3e, Supplementary Fig. 17a, c). With the increasing spike $n_S$, the difference $\Delta$aEPSC between aEPSC peak value and quiescent current increased monotonically due to the enhanced serial migration of cations (Fig. 3f). When $n_S = 50$, the aEPSCs in ZTO-7:3 and ZTO-1:1 ASs decayed to the initial value in a short time; i.e., they showed short-term plasticity (STP). In contrast, when $n_S = 50$ for ZTO-3:7 AS, the aEPSC decay took > 1 min; i.e., it shows long-term plasticity (LTP). These results show that the plasticity can be easily tuned between STP and LTP by adjusting the Zn:Sn molar ratio.

Biological synapses can act as filters to process dynamic real-time information[47,48]. To verify the high-pass filtering characteristics of the three ZTO ASs, the spike-frequency-dependent plasticity (SFDP) was examined by applying consecutive spikes (2 V, 50 ms) with different spike frequencies ($f_S = 0.5, 1, 2, 2.5, 5, 10$ Hz). With increasing $f_S$, the aEPSC of three ZTO ASs increased monotonically because the migration of cations was increased and their back-diffusion was suppressed (Fig. 3g, Supplementary Fig. 17b, d). aEPSC gain was defined as $G_{aEPSC}$ [%] $= 100 \cdot A10/A1$, where A$n$ is the aEPSC of the $n$th peak. $G_{aEPSC}$ increased as the $f_S$ increased from 0.5 to 10 Hz (Fig. 3h); this result demonstrates that the devices have high-pass filtering characteristics. $G_{aEPSC}$ was highest in ZTO-3:7 AS, because it has the largest number of ion-trap sites.

Spike-duration-dependent plasticity (SDDP) and spike-voltage-dependent plasticity (SVDP) are essential rules in synaptic learning and memory[49,50]. SDDP of the three ZTO ASs was obtained by adjusting duration $d_S$ (from 50 to 500 ms) of presynaptic spikes (2 V, 50 ms) (Fig. 3i). As $d_S$ increased, the number of cations migrations increased, so $\Delta$aEPSC increased in all three devices (Fig. 3j). The aEPSC peak of the three devices also increased as the amplitude of presynaptic spikes increased. The responses after repeated stimuli differed among the ASs. The ZTO-3:7 AS retained the aEPSC after intensive stimuli (Fig. 3k), and therefore displayed LTP. However, the aEPSC of ZTO-7:3 and ZTO-1:1 ASs decayed to the initial value in a short time (Supplementary Fig. 18), so they only possessed STP.

Furthermore, we demonstrated the potential applications of ZTO-fibers ASs in encrypted communication and Pavlovian associative learning. Owing to their STP, ZTO-7:3 and ZTO-1:1 ASs can be used to generate international Morse code which is composed of "dot (.)" and "dash (−)"[51,52]. For ZTO-1:1 AS, the output currents smaller than the threshold (7.5 μA) were defined as "dot (.)", which were triggered by a short-duration spike (2 V, 50 ms), and output currents bigger than the threshold were defined as "dash (−)", and were triggered by a long-duration spike (2 V, 150 ms). The input letters "NKU" and "YES" were successfully generated using ZTO-1:1 AS (Fig. 3l). For ZTO-7:3 AS (the threshold set as 1.5 μA), the input letters "NANO" and "NKU" were successfully represented (Supplementary Fig. 19).

In addition, the LTP of ZTO-3:7 AS was exploited to simulate Pavlovian associative learning (Fig. 3m). Two groups of ten consecutive spikes (1 V, 50 ms) and (2 V, 50 ms) were applied as the "bell" and "bone", respectively. An aEPSC = 14.5 μA was defined as the threshold for the salivation response. Before training, ten consecutive ringing "bell" stimuli did not induce salivation. Then, salivation was evoked by ten consecutive "bone" stimuli. During training, the "bell" stimuli and "bone" stimuli were alternately applied on the device, and the consequent salivation can be observed. After training, a series of "bell" stimuli were applied and induced salivation; this result indicates that Pavlovian associative learning was successfully simulated using ZTO-3:7 AS.

The flexible and stretchable properties of artificial synapses are of great significance for wearable and human-interface applications[53–56], neuromorphic computing for artificial intelligence[57,58], especially on-body data processing[59]. Thus, we fabricated a flexible ZTO-3:7 AS on a polyimide (PI) substrate and investigated its flexible characteristics. The synaptic properties of the device show excellent bending resistance (<7% variation after 2000 times bending), so it has potential applications in flexible electronics (Supplementary Fig. 20).

## An artificially-intelligent cornea with tactile sensation

The corneal reflex can be divided into the bilateral reflex in which both eyes blink, and the unilateral reflex in which only one eye blinks. Furthermore, the unilateral reflex can be divided into the ipsilateral reflex, which is closure of the eye on the same side as the stimulus, and the contralateral reflex, which is closure of the eye on the side opposite to the stimulus[60]. These variants of the reflex use different neural pathways. In a neurological examination, the corneal reflex is evoked by using a cotton wisp to touch the cornea[61,62] (Fig. 4a). The value of corneal reflex examination lies in confirming the integrity of the neural pathway[63]. During the bilateral corneal reflex (Fig. 4b) the mechanoreceptors within the cornea act as the receptors of the reflex. Sensory information transmits to the spinal trigeminal nucleus through the trigeminal nerve (V). Then, the information is transmitted to the bilateral facial nucleus from the spinal trigeminal nucleus through nerves between them. Finally, the efferent information is transmitted to the bilateral orbicularis oculi muscles (as actuators) through the facial nerve (VII) and induces its contraction. Dysfunction of the corneal reflex can be caused by damage to the central or peripheral nervous system in the pathway[64]. Physicians can determine the origin of some diseases by examining the corneal reflex.

Herein, we constructed an artificial corneal reflex arc that contains a vibration sensor-oscillation circuit (Supplementary Fig. 21a, b), a ZTO-3:7 AS, an amplifier circuit, and a pair of electrochromic devices (Fig. 4c, Supplementary Fig. 22). The vibration sensor-oscillation circuit acts as a receptor to convert mechanical stimuli into electrical pulses, the ZTO-3:7 AS acts as the processing core to transfer and integrate the information, the amplifier circuit outputs the desired voltage to operate the actuator, and the electrochromic device acts as the actuator that responds to the postsynaptic current. As the processing core, the ZTO-3:7 AS was selected due to its excellent synaptic activity. The contraction of the orbicularis oculi muscles decreases the light intensity that enters the eyes. Therefore, the electrochromic device, which can change from light blue to dark blue and further decrease the transmittance, was selected as an actuator to mimic the contraction of the orbicularis oculi muscles.

The sandwich-type electrochromic device composes of ITO/PET substrates, an electrochromic layer that uses PEDOT: PSS, and a gel electrolyte layer that uses LiClO$_4$ (Fig. 4d). The color change of PEDOT: PSS is caused by the reversible insertion/extraction of Li$^+$ under different stimulation voltages[65,66]. The PEDOT: PSS could react with ions during color changing according to the following redox mechanism: (PEDOT$^{x+}$PSS$^{x-}$) + xLi$^+$ + xe$^-$ ↔ x(PEDOT$^0$PSS·Li$^+$). As stimulation voltages were increased from 0 V to 3 V, the color of the electrochromic device

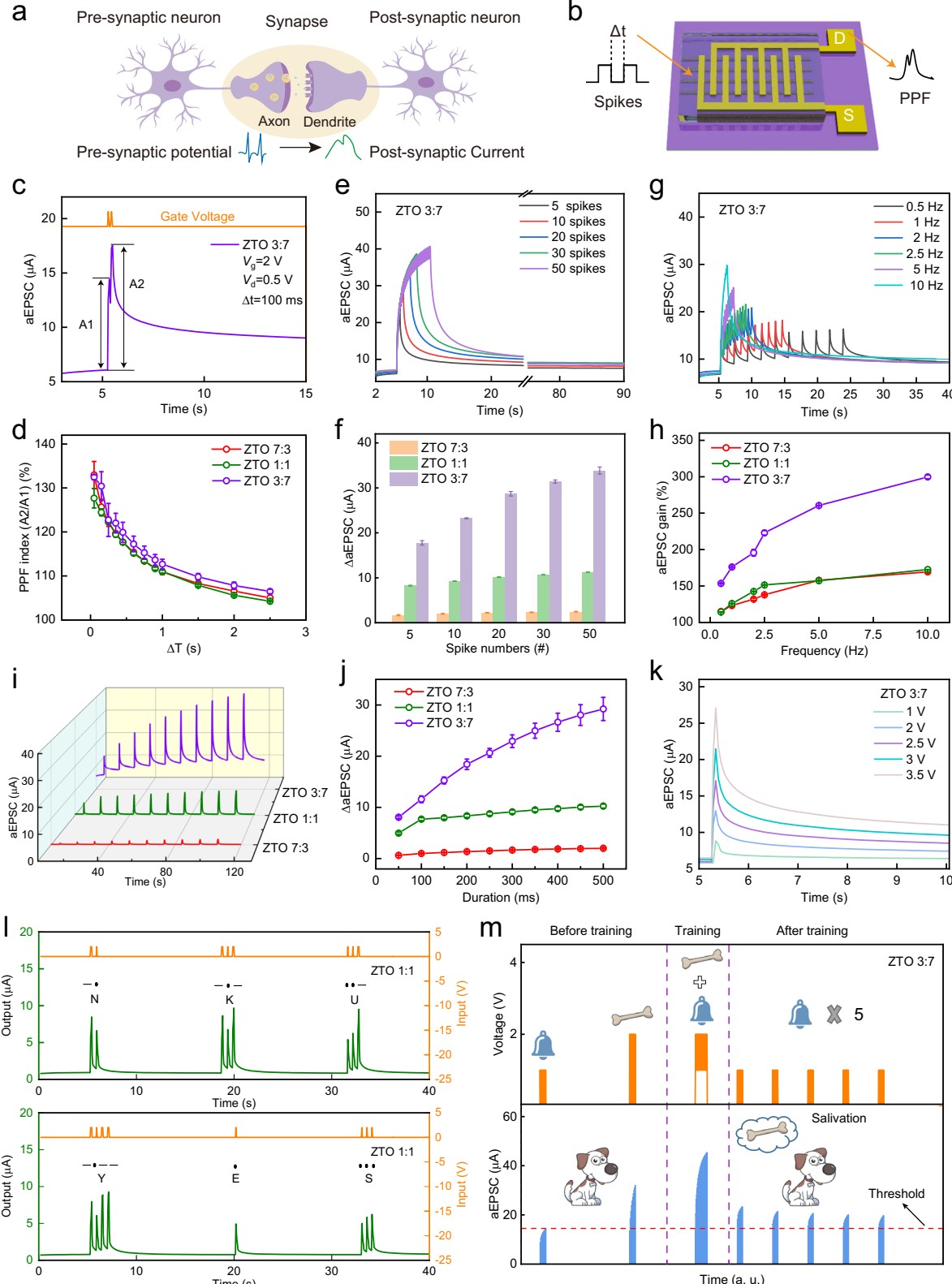

**Fig. 3 | ZTO-fibers artificial synapses with tunable synaptic plasticity.**
**a**, **b** Schematic illustration of synaptic signal conduction under a pair of stimuli: biological synapse (**a**) and ZTO-fibers AS (**b**). **c** aEPSC of ZTO-3:7 AS evoked by a pair of spikes. **d** PPF index versus spike interval for three ZTO ASs with different Zn:Sn molar ratios. **e**, **f** aEPSC of ZTO-3:7 AS (**e**) and ΔaEPSC of three ZTO ASs with different Zn:Sn molar ratios (**f**) triggered by consecutive spikes with different spike numbers. **g** aEPSC of ZTO-3:7 AS triggered by consecutive spikes with different

spike frequencies. **h** aEPSC gain versus frequency for three ZTO ASs with different Zn:Sn molar ratios. **i**, **j** aEPSC (**i**) and ΔaEPSC (**j**) triggered by spikes with different spike duration for three ZTO ASs with different Zn:Sn molar ratios. **k** aEPSC of ZTO-3:7 AS triggered by spikes with different amplitude. **l** International Morse code of "NKU" and "YES" realized using ZTO-1:1 AS. **m** Pavlovian learning behavior realized using ZTO-3:7 AS.

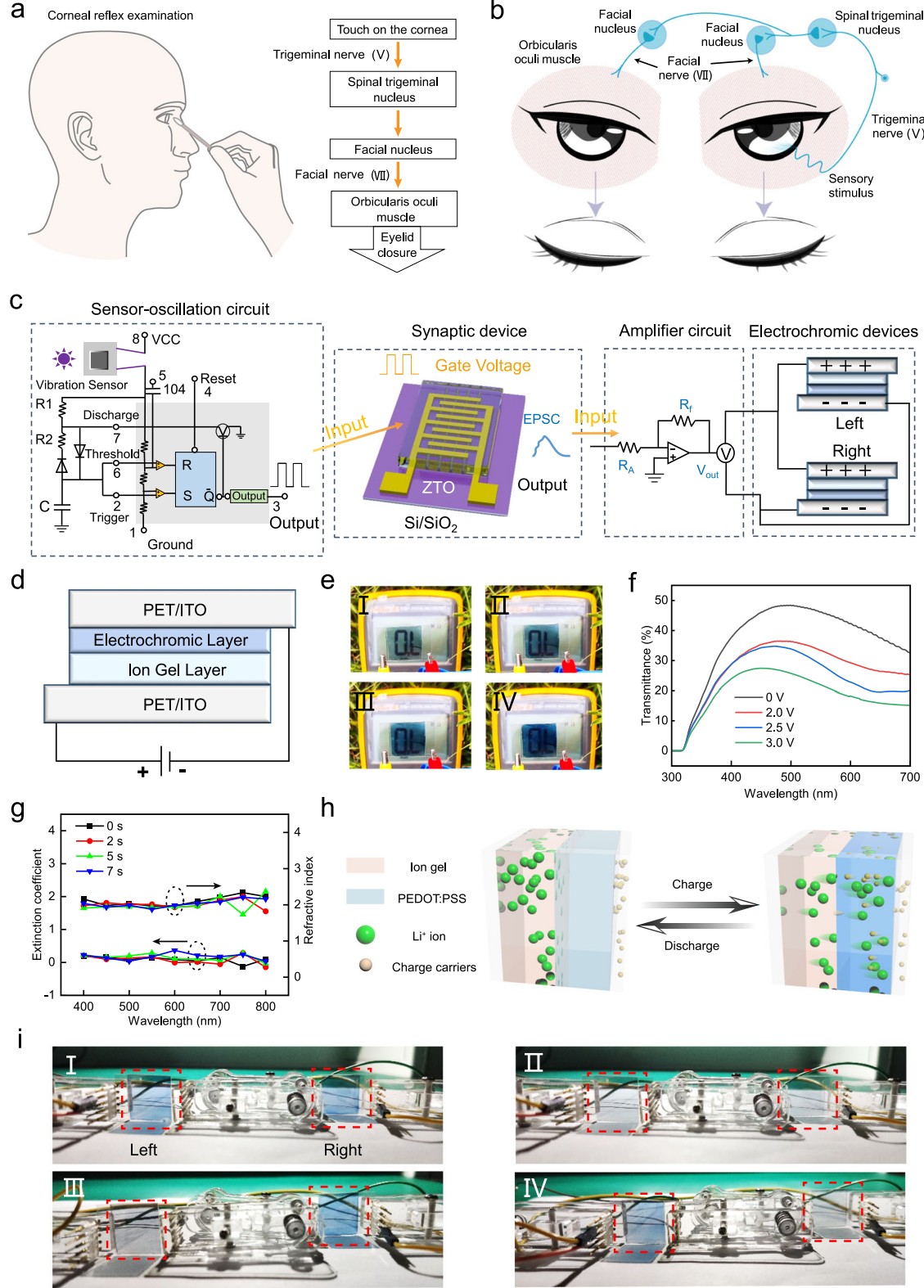

**Fig. 4 | An artificially-intelligent cornea with tactile sensation. a** Schematic of corneal reflex examination (left), and the process of corneal reflex (right). **b** Schematic of corneal reflex evoked by mechanical stimuli. **c** Configuration of artificial corneal reflex arc responding to mechanical stimuli. **d** Schematic of the structure of the electrochromic device. **e, f** Digital images (**e**) and optical transmittance spectra (**f**) of the electrochromic device under different stimulation voltages. **g** Refractive index and extinction coefficient of the electrochromic device under different stimulation time. **h** Schematic illustration of the Li⁺ ions and charge transportation through the electrochromic device. **i** Digital images of electrochromic actuators under bilateral reflex (I), without reflex (II), ipsilateral reflex (III), and contralateral reflex (IV).

gradually darkened (Fig. 4e), and consequently its transmittance decreased (Fig. 4f). The response time of the electrochromic device, defined as the period to achieve a 90% change in transmittance, is ~4.3 s under 3 V (Supplementary Fig. 23a, b). Considering the duration of eyelid closure is approximately 100 ~ 150 ms[67], the change in transmittance of the electrochromic device in 150 ms was calculated as ~9% (Supplementary Fig. 23c). Thus, the response speed of the device can partially replicate the eyelid-closure process. In the visible range under stimulation durations of 0, 2, 5 and 7 s, the average refractive index $n$ of the electrochromic device was ~2.04, and the extinction coefficient $k$ was ~0.125 (Fig. 4 g). These results demonstrate that the device has refractive capacity.

The statistical curve of signal output over time for each part of the artificial corneal reflex arc under mechanical stimulation was determined (Supplementary Fig. 24). When a finger or a foreign object touches the vibration sensor, it acts as a normally closed device that conducts at this time, making the power-supply side of the oscillator circuit turn on and output a high-frequency spikes pulse. The output spike signals are used as the presynaptic stimulus to trigger the post-synaptic current of the artificial synapse, then an amplifier circuit outputs a voltage signal, which is used as the modulation signal of the electrochromic unit. Gradual changes of electrochromic actuators under the real-time touch can be seen in Supplementary Movie 1. When the artificial corneal reflex arc was evoked by mechanical stimuli, the $Li^+$ ions and charge migrated through the electrochromic device (Fig. 4h); as a result, the color of the electrochromic actuator changed to dark blue. Under the same vibration stimulus, the different states of electrochromic actuators can represent different damage points of the neural pathways (Supplementary Table 4). When the stimulus was applied to the right eye (Fig. 4i, Supplementary Fig. 25, Supplementary Movie 2), all parts of the system operated normally, and both electrochromic actuators changed to dark blue (i.e., the bilateral reflex); this response corresponds to a corneal reflex arc without dysfunction (Fig. 4i I). Afferent pathway damage or central dysfunctions cause deficits in corneal reflex bilaterally; in this situation, the afferent pathway or synaptic device in the system was disconnected, and the electrochromic actuators remained light blue (Fig. 4i II). Contralateral and ipsilateral efferent damages correspond to ipsilateral and contralateral reflex, respectively. When the left efferent pathway between the synaptic device and electrochromic actuators was disconnected, the right electrochromic actuator changed to dark blue (Fig. 4i III). In contrast, when the right efferent pathway between the synaptic device and electrochromic actuators was disconnected, the left electrochromic actuator changed to dark blue (Fig. 4i IV). These results demonstrate that the artificial corneal reflex arc was successfully constructed, and can be used to simulate the neurological examination.

The artificial corneal reflex arc has protection, tactile perception, and light refraction functions like the native human cornea, and it can also mimic the contraction of the orbicularis oculi muscles. It is more intelligent than the native human cornea and the conventional artificial corneas that do not perceive tactile stimuli. Thus, we describe our artificial corneal reflex arc as an artificially-intelligent cornea with tactile sensation.

### Sensory expansion and interactive functions of the artificially-intelligent cornea

A natural cornea does not assume the functions of light perception and regulation of the changes in light levels. Light enters the eyes through the cornea, and is detected by the retinal photoreceptors (rods and cones), then the elongated photoreceptors convert the light signal to action potentials, which are transmitted to the brain through the optic nerve (Fig. 5a). To increase the smartness of our artificially-intelligent cornea with tactile sensation, we endowed it with light-perception ability by using a light sensor-oscillation circuit

(Supplementary Fig. 21c, d) to replace the vibration sensor-oscillation circuit (Fig. 5c). The artificially-intelligent cornea realized sensory expansion, and obtained the ability to perceive and respond to external light stimuli.

With the light illuminance increasing from 22 to 583 lx, the frequency of presynaptic spikes (the electric pulses output by the light sensor-oscillation circuit) increased from 0.5 Hz to 10 Hz (Fig. 5d). The statistical curve of signal output over time for each part of the artificially-intelligent cornea under light stimulation was also determined (Fig. 5e). The aEPSC of ZTO-3:7 AS triggered by different light illuminances induced the electrochromic actuator to remain light blue under 22 lx, and to change to dark blue under 583 lx. These results demonstrated that the opening and closing state of human eyes were successfully imitated by an electrochromic actuator, which can realize the regulation of transmitted light and protection for human eyes (Fig. 5f, g). Furthermore, the transmittance of the electrochromic actuator can adapt to changes in the light intensity in the environment (Fig. 4f, Supplementary Movie 3). As light intensity increased, the frequency of presynaptic spikes increased, so the aEPSC increased and as a result, the transmittance of the electrochromic actuator decreased gradually. The light-adaption of transmittance demonstrated that the artificially-intelligent cornea can interact with the environment, to supply additional adaptive protection for the eyes in the changing environment. Therefore, the artificially-intelligent cornea simultaneously realized sensory expansion and interactive functions.

As a proof-of-concept, a robot was equipped with the artificially-intelligent cornea (Fig. 5b), which was light blue with high transmittance under weak light, and changed to dark blue with low transmittance under bright light (Fig. 5h). Notably, the artificially-intelligent cornea is logically compatible with the human nervous system, replicates protection, tactile perception, and light refraction functions of the native human cornea, and expands light-perception and interactive functions. In the future, the development tendency of the novel artificially-intelligent cornea will be oriented towards biocompatibility, stability, miniaturization, and high integration. After development and optimization, the artificially-intelligent cornea may be compatible with transplantation into individuals who have corneal blindness and are waiting for medical interventions, thereby alleviating the shortage of donor corneas. We believe that the mature artificially-intelligent cornea has potential applications in neuroprosthetics and visual rehabilitation.

## Discussion
In summary, we have developed an artificially-intelligent cornea that replicates the functions of the native human cornea such as protection, tactile perception, and light refraction, and expands light perception and interactive functions. To realize these functions, an artificial corneal reflex arc that can implement mechanical and light information coding, information processing, and the regulation of transmitted light was constructed by integrating sensor-oscillation circuits, ZTO ASs, and electrochromic devices. Heavy-metal-free ZTO, which is nontoxic and low-cost, was fabricated as digitally-aligned, long and continuous fabric patterns with high transmittance (99.89%) and low haze (0.36%) to ensure the optical properties of the artificial synapse. By tuning the crystal phase structures of ZTO fibers, the tunable synaptic plasticity between STP and LTP can be easily realized, and further used in encrypted communication and associative learning. In the future, the biocompatibility, stability, size, and integration degree of the artificially-intelligent cornea must be optimized. The mature artificially-intelligent cornea may have applications in neuroprosthetics and visual restoration.

## Methods
The Supplementary Information displays additional experimental details.

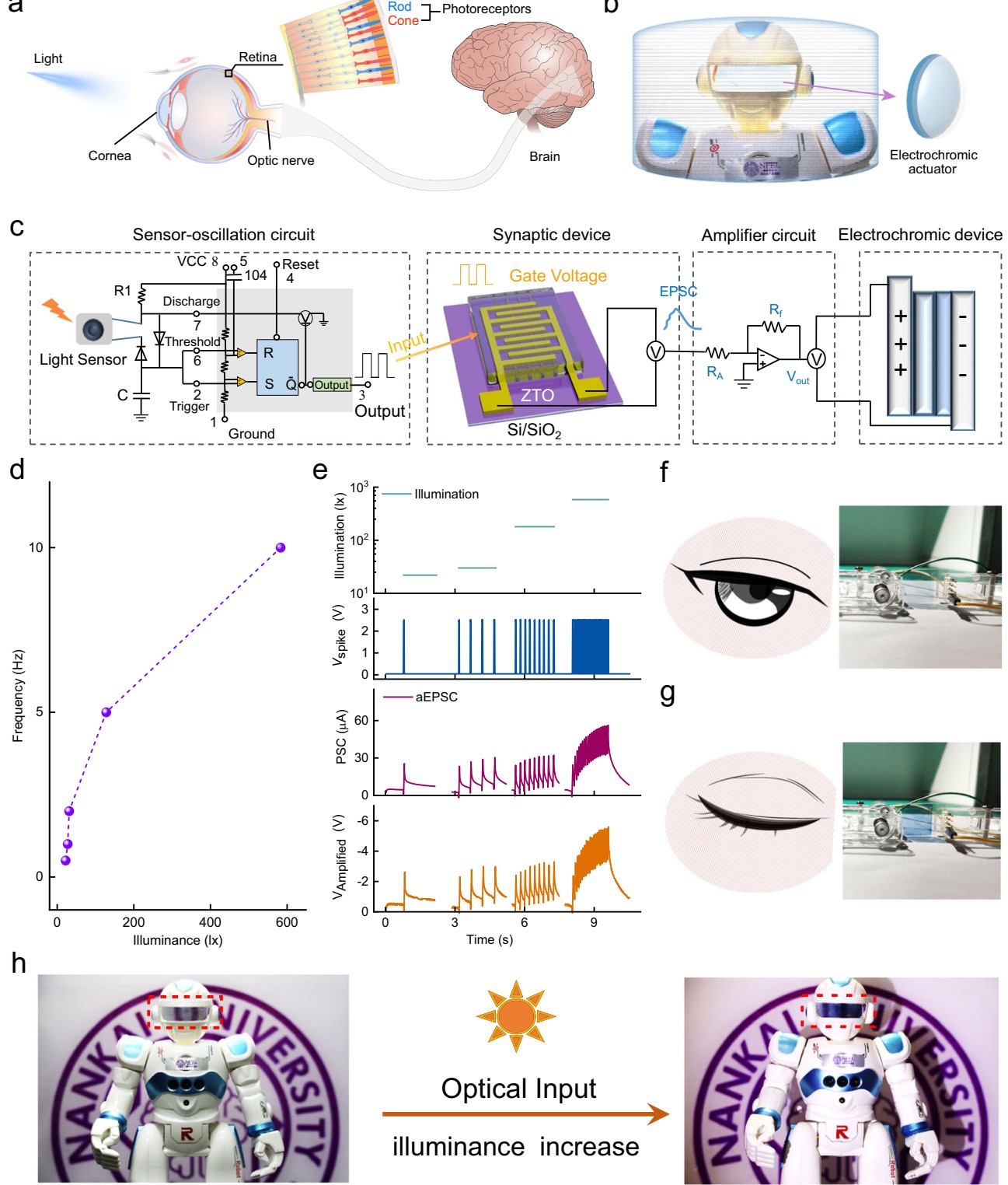

**Fig. 5 | Sensory expansion and interactive functions of the artificially-intelligent cornea. a** Schematic of the light perception in the human eyes. **b** Schematic of a robot equipped with an artificially-intelligent cornea. **c** Configuration of the artificial corneal reflex arc responding to light stimuli. **d** Frequency of the output spikes from light sensor-oscillation circuit versus illuminance. **e** Statistical curve of signal output over time for each part of the artificially-intelligent cornea under light stimulation. **f, g** Digital images of an electrochromic actuator and the corresponding schematic of the human eye under weak light (**f**) and bright light (**g**). **h** Digital images of a robot equipped with an artificially-intelligent cornea.

## Materials

Zinc nitrate hexahydrate ($Zn(NO_3)_2 \cdot 6H_2O$, 98%), stannous chloride dehydrate ($SnCl_2 \cdot 2H_2O$, 98%), and Poly(vinylpyrrolidone) (PVP, Mw = 1,300,000 g mol$^{-1}$) were purchased from Acros, Innochem, and Rhawn, respectively. *N,N*-Dimethylformamide (DMF, 99.8%) was purchased from Meryer. The 4-inch Si/SiO$_2$ wafers (~500 μm thick) with a SiO$_2$ layer of ~300 nm were purchased from Tebo Technology Co., Ltd. Electrochromic ink (CHRO-EP202-P) and gel electrolyte (ENER-EI30M)

were purchased from Shanghai Mifang Electronic Technology co. LTD. All these reagents were used without further purification.

## Printing ink preparation

The total concentration of the metal precursor in the solution was fixed, and the molar ratio of Zn:Sn was fixed to 7:3, 1:1 or 3:7, as appropriate. $Zn(NO_3)_2 \cdot 6H_2O$, $SnCl_2 \cdot 2H_2O$ and PVP were dissolved in DMF and then stirred at 50 °C for 12 h to form an ink.

## ZTO fibers fabrication

Controllably-aligned ZTO fibers were digitally printed on a $Si/SiO_2$ substrate by using an electrohydrodynamic nanowire printer according to the following parameters. The ink was pumped into an injection syringe and injected downward through a metallic nozzle tip. For this purpose, a high voltage of about 1 kV was applied to the nozzle, while the nozzle tip-to-collector distance was maintained at 3.5 mm. The ink was injected through a metal nozzle at an injection rate of 50 nL/min. After being printed, the samples were annealed in a muffle furnace at 500 °C for 2 h.

## ZTO fibers-based artificial synapse fabrication

A $Si/SiO_2$ substrate was ultrasonically cleaned in deionized water, isopropanol, acetone and anhydrous ethanol for 30 min, sequentially. Then digitally-aligned ZTO fibers were printed on the substrate surface to produce the channel. The printed ZTO fibers were annealed in a muffle furnace at 500 °C for 2 h. Gold source and drain electrodes (80 nm) were then thermally deposited through a shadow mask. Ion gel was obtained by mixing poly(vinylidene fluoride-co-hexafluoropropylene) (PVDF-HFP), 1-ethyl-3-methylimidazolium bis-(trifluoromethylsulfonyl)imide ([EMIM-TFSI]), and acetone in a weight ratio of 1:4:7 at room temperature, then drying the solution in a vacuum oven at 70 °C for 30 min. Finally, the prepared ion gel was transferred onto the ZTO fibers channel area. A metal probe that contacted the ion gel was used as the presynaptic input terminal to apply the presynaptic spikes. For an artificial synapse, there are ~19 fibers and correspondingly ~288 Au/ZTO junctions engaged in the formation of postsynaptic current.

## Data availability

The data generated in this study are provided in Supplementary Information and Source Data file, or from the corresponding author upon reasonable request. Source data are provided with this paper.

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

## Acknowledgements

This work was supported by the National Science Fund for Distinguished Young Scholars of China under grant no. T2125005 (W.X.), the National Key R&D Program of China under grant no. 2022YFE0198200, 2022YFA1204500, and 2022YFA1204504 (W.X.), the Tianjin Science Foundation for Distinguished Young Scholars under grant no. 19JCJQJC61000 (W.X.), the Shenzhen Science and Technology Project under grant no. JCYJ20210324121002008 (W.X.), the National Natural Science Foundation of China under grant no. 62204131 (L.S.), and the China Postdoctoral Science Foundation under grant no. 2023T160336 (L.S.).

## Author contributions

W.X. conceived and designed the research. W.X., S.Q., and L.S. designed the systems. S.Q. performed the experiments. S.Q. conducted the characterization and measurements. L.S. designed and prepared the circuits. W.X., S.Q., and L.S. contributed to analysis and discussion on the data. S.Z., Ji.L., Y.L., and Ju.L. helped improved data visualization. S.Q. and W.X. wrote the manuscript. All authors discussed the results and commented on the manuscript.

## Competing interests

The authors declare no competing interests.
