## [Peer Review File · Nature Communications]

REVIEWER COMMENTS

Reviewer #1 (Remarks to the Author):

This is an interesting paper. The authors introduce an artificial cornea that could assume several of the functions of the native human cornea and may help restore vision in people with corneal blindness. However, there is incorrect information, and the conclusions are not supported by the results, especially in the fields of rehabilitation and visual neuroprostheses.

General comments:

- Researchers should be aware that light perception is not a function of the cornea. The abstract and introduction should be rewritten.
- The authors should provide more recent references regarding corneal transplantation and keratoplasty.
- The cornea has not “feelings”. It does contain a large number of sensory nerve fibers that are responsible to transmit important sensations to the brain, such as touch, pressure, and temperature.
- Figure 2A is cropped and does not show a representative sample of a biological synapse.
- Although several studies have shown the potential use of ZTO to create devices that could mimic the function of biological synapses, the term excitatory postsynaptic currents (EPSCs) should be restricted to recordings related to the release of neurotransmitters from presynaptic neurons. Perhaps a more suitable name could be: “Artificial Excitatory Postsynaptic Currents (aEPSCs)”.
- Information about the rise time of the artificial synapses should be provided.
- The number of experiments performed regarding synaptic plasticity is unclear. More information should be provided. Furthermore, Figure 3 should show mean values and deviation measures.
- Figure 4A is misleading. The optic nerve is misplaced.
- Although the cornea plays an important role in regulating the amount of light that enters the eye, it does not respond directly to changes in light levels; therefore, the conceptual design of the proposed artificial cornea is unclear. Additionally, the conceptual design and fabrication of the proposed approach in the biomedical field is missing.
- As stated before, the cornea does not have any photoreceptor cells that detect light levels. Therefore, the schematic of the corneal reflex triggered by bright light (Fig 5) is misleading.
- The conclusions are not supported by the results.

- Discussion should include possible limitations of the study.
- There are several typos, such as "A stimulation voltages were increased... (pag#15). English should be reviewed.

Reviewer #2 (Remarks to the Author):

In the paper authored by Qu et al., a novel device and system design is presented to emulate the reflex reactions of the cornea in closing eyelids. The system comprises four modules: a sensor-oscillator circuit, a ZTO-nanofiber-based synaptic device, an amplified circuit, and an electrochromic device. The application of the electrochromic device to mimic eyelid closing functionality is particularly intriguing. The ZTO nanofiber provides excellent transparency and tunable performance for the synaptic device, allowing for leaky integration of the pulsed sensor output. This work is anticipated to inspire further advancements in the application of neuromorphic computing concepts to artificial eye system development. To further consider this work for publication, the authors should address the following comments:

1. Comprehensive performance characterization results for both touch and optical sensor-oscillator circuits are required, particularly with measurements of circuit output under gradually varied pressure or light intensity.
2. The paper should include temporal response information for the electrochromic device and discuss the extent to which this response speed can replicate the eyelid closure process.
3. To assess the operation of the demonstrated artificial corneal system, signal measurements at each stage of the signal processing path (e.g., after the synaptic device and amplification) are necessary.
4. The authors should elaborate on how this system can achieve the four distinct reflex types mentioned in the paper and depicted in Fig. 4k.
5. In addition to ZTO transparency, it is essential to measure the transparency of the entire synaptic device.
6. To facilitate audience comprehension, the authors should consider including videos that demonstrate the real-time touch/light-induced switching of the electrochromic devices in conjunction with figures 4k, 5g, and 5h.
7. As the ultimate objective is to integrate such systems with patients or even soft robots, the importance of soft and stretchable properties cannot be understated. The authors are encouraged

to address this aspect in the paper, referring to recent works on developing stretchable neuromorphic devices for human-interfaced applications.

Reviewer #3 (Remarks to the Author):

This research has brought a novel concept in design of artificial cornea with the so-called feeling sensibility. The idea of tailoring an electro-chromatic system into an integrated system for development of a light-sensitive nociceptive system is interesting. However, the present style of manuscript needs considerable improvement to be considered as a high-quality research work that fulfills the expected standards. Generally, you should provide enough details on the methods and working mechanism of your system. The work is not reproducible in the present style. Authors are invited to go through the following recommendations for improving the quality of their research work via major revision:

1- The title of manuscript is not informative. It is necessary to include more information in title to reflect the mechanism of performance of this system.

2- In introduction section "It may help restore the vision in individuals who have corneal blindness" This is a powerful claim that is hard to be grasped from the findings of manuscript. You may use the similar expressions in conclusion, but this is not practically proved.

3- The introduction section needs considerable alteration and improvement. The present style is not informative and cannot satisfy the quality of a well-structured manuscript. Authors are invited to alter their approach toward the presentation of proposed challenges, the importance of their innovative idea, and their strategy to tackle the challenge. They may also go through different research papers that proposed different approaches and mechanisms for design of artificial cornea. Artificial cornea have developed previously, please explain the previous achievements and then elaborate your approach and solving strategy. Will the work be of significance to the field and related fields? How does it compare to the previous published works?

4- Authors are recommended to label 3 different components of their system in Figure 1, i.e. the sensor-oscillation circuit, processing core, actuator.

5- It is necessary to provide the statistics information about the number, and coverage area of surface with ZTO fibers. We should know how many ZTO fibers are deposited, and then determine what the contact area is between the ionic gel and the ZTO fibers.

6- Furthermore, in lines 437-446, it was claimed that ZTO fibers were printed on the Si/SiO₂ substrate surface, followed by fabrication of Au electrodes, and then transfer of ion gel. We suppose that SiO₂ is developed on Si. Please clarify and explain the method of deposition of SiO₂ and its possible thickness. Therefore, according to the method section, the structure is "Si/SiO₂-ZTO

fiber/Au electrode/Ion gel". But, if authors return to the graphical image of synaptic device, it shows that Au electrodes are deposited on Si/SiO₂ substrate, followed by development of ZTO fiber arrays on top of Au electrode. Therefore, authors should provide the correct graphical scheme of their ZTO fiber based artificial synaptic system in Figure 1, Figure 2, Figure 3, and Figure 4.

7- Moreover, if the Au electrodes are developed on top of the ZTO fibers, authors should provide a clear information that explain how many Au/ZTO junctions are engaged in the formation of postsynaptic current, and also elaborate the way that pre-synaptic current is applied. Generally, the materials and method section should clearly explain the details of fabrication and measurement process.

8- In Figure 2g, authors should provide the SAED pattern or high-resolution fast Fourier-transform (FFT) pattern of ZTO crystalline structure to confirm their points.

9- Figure 2l is not readable. Either provide a clear version or transfer the file into the supplementary information section.

10- Line 132...please edit the sentence. Was deconvoluted.

11- What does PEDOT:PPS represent?

12- How did you calculate the gate capacitance in Note 1?

13- It was explained that the structure of ZTO 1:1 fibers was disordered after simulated annealing? It is the first time and the last time that manuscript talk about annealing? Is there any annealing process in fabrication of ZTO fibers? Please clarify this issue.

14- It is necessary to practically measure the bandgap of ZTO fibers and compare it with the result of DOS in Figure 2n and Figure S 11.

15- Please elaborate how the mechanical stimuli turn on the vibration sensors, including the frequency of vibration and its information.

16- In Figure 5f, a possible application of the bio-compatible artificial cornea is explained, however, there is no evidence or outcome about this function. Therefore, there is no point to allocate a figure to a concept which is not developed.

17- I strongly recommend to add video files to shows the performance of artificial cornea with feelings. You may provide several brief video files that shows the gradual changers of electro-chromic actuators at different intensity of optical inputs.

Response to Reviewers' Comments

(Manuscript ID: NCOMMS-23-17850)

We thank the editor and reviewers for considering our manuscript. We sincerely appreciate your valuable comments and constructive suggestions. These remarks are all valuable and helpful for improving the quality of our work.

In our revision, we have thoroughly modified the content of the manuscript according to the reviewers' comments and suggestions. We have modified Figures 1-5 in the revised manuscript and provided more technical details. We have added 12 figures, 1 note, and 1 table in the revised manuscript and Supplementary Information and provided 3 brief video files to support our responses to the comments.

In the following, we present a detailed response to all the comments raised by the reviewers point-by-point. Please find below our responses (in blue) to each of your comments (in black). Revisions to the original manuscript and Supplementary Information are highlighted in red.

Reviewer #1:

This is an interesting paper. The authors introduce an artificial cornea that could assume several of the functions of the native human cornea and may help restore vision in people with corneal blindness. However, there is incorrect information, and the conclusions are not supported by the results, especially in the fields of rehabilitation and visual neuroprostheses.

Response: We would like to thank the reviewer for the positive comments and the valuable suggestions on our work. We have revised the manuscript according to each of your specific comments. The point-to-point responses to the reviewer's comments are presented below.

Comment 1 from Reviewer #1:

Researchers should be aware that light perception is not a function of the cornea. The abstract and introduction should be rewritten.

Response: Thank you very much for your valuable comment. We feel sorry for the confusion. We have modified the corresponding content in the revised manuscript. In response to your comment, we have rewritten the abstract and introduction, and cited additional references. The reviewer may refer to the following content in the revised manuscript:

Abstract: "We demonstrate an **artificially-intelligent** cornea that can assume the functions of the native human cornea such as protection, **tactile perception**, and **light refraction**, and **possesses sensory expansion and interactive functions**. These functions are realized by an artificial corneal reflex arc that is constructed to implement

mechanical and light information coding, information processing, and the regulation of transmitted light. Digitally-aligned, long and continuous zinc tin oxide (ZTO) semiconductor fabric patterns were fabricated as the active channels of the **artificial synapse**, which are non-toxic, heavy-metal-free, low-cost, and ensure superior comprehensive optical properties (transmittance > 99.89%, haze < 0.36%). Precisely-tuned crystal-phase structures of the ZTO fibers enabled reconfigurable synaptic plasticity, which is applicable to encrypted communication and associative learning. This work suggests new strategies for the tuning of synaptic plasticity and the design of visual neuroprosthetics, and **has important implications for the development of neuromorphic electronics and for visual restoration.**”

Introduction:

“Located at the front of the eye, a cornea is a transparent structure that provides focusing power and shields the iris and lens from foreign substances¹. It is the most densely-innervated part of the body, **and is therefore** sensitive to foreign contaminants; a touch of the cornea causes an involuntary reflex to close the eyelid (the corneal reflex)². However, corneal diseases can result in blindness, and more than 10 million individuals worldwide suffer from bilateral corneal blindness³⁻⁵. **Corneal transplantation of a donor allograft (i.e., keratoplasty)** is the most common medical solution to this **disease**⁶⁻⁸, but due to scarcity of donated corneas, it can be used on only 1 in 70 patients⁹⁻¹¹, and 12.7 million patients are waiting for the process^{12,13}.

As a solution to this scarcity, artificial corneal substitutes that comply with the requirements for optical and transparency have been developed¹⁴. The Boston keratoprosthesis (Kpro) is the most-commonly implanted artificial corneal substitute worldwide; it is a collar-button-shaped device that is mainly made of polymethyl methacrylate (PMMA)¹⁵⁻¹⁷. Aurolab Kpro shares the design with Boston Kpro and is considered an alternative to the latter when affordability is a limiting factor¹⁸. MICO F Kpro contains a central ring with a threaded PMMA optic that is supported by a titanium frame, and the surgical invasiveness is mild¹⁹⁻²¹. As a unique approach to the artificial cornea, the osteo-odonto-keratoprosthesis is a biological Kpro that uses the lamina derived from an autologous mono-radicular tooth as a frame for PMMA optical cylinder^{22,23}. All of these types of artificial cornea have been applied clinically to improve vision in cases of severe corneal opacification, and the effects of visual restoration, long-term safety, practicality, and cost are acceptable to the patients.

Although existing artificial corneas can assume partial functions of the native human cornea, such as protection and light refraction, they do not reconstruct the tactile sensation and therefore do not realize the corneal reflex. Thus, trying to develop a ‘smarter’ artificial cornea with tactile sensation and even possessing sensory expansion and interactive functions is of great significance for restoring vision in corneal blindness. Moreover, these functions must be realized by ‘invisible’ built-in

electronics²⁴⁻²⁸, to achieve the high transparency and low haze of a natural cornea.

This paper describes an **artificially-intelligent** cornea that has the functions of protection, tactile perception and light refraction **like the native human cornea, and also has sensory expansion (widening an existing sensory experience) and interactive functions**. The **touch sensitivity** of the cornea was reconstructed by a reflex arc composed of sensor-oscillation circuits, zinc tin oxide (ZTO) artificial synapses (ASs) and electrochromic devices, which implement mechanical and light information coding, information processing and the regulation of transmitted light. Heavy-metal-free ZTO fabric patterns were fabricated with long, continuous morphology and digital alignment, and used as active channels due to their non-toxicity, low-cost, high transmittance (> 99.89%), and low haze (< 0.36%) to ensure the optical properties of the **artificial synapse**. The crystal phase of the ZTO fibers was highly tuned, and therefore ensured tunable synaptic plasticity, and incidental applications to encrypted communication and associative learning. **This work presents a new resource that may be used in development of powerful neuromorphic electronics and visual neuroprosthetics.”**

Reference:

6. Bennett, V. G., Alberti, M., Quadrio, M. & Pralits, J. O. Optimization of patient positioning for improved healing after corneal transplantation. *J. Biomech.* **150**, 111510 (2023).
7. Lemaitre, D., Tourabaly, M., Borderie, V. & Dechartres, A. Long-term outcomes after lamellar endothelial keratoplasty compared with penetrating keratoplasty for corneal endothelial dysfunction: a systematic review. *Cornea* **42**, 917-928 (2023).
8. Viberg, A., Vicente, A., Samolov, B., Hjortdal, J. & Bystrom, B. Corneal transplantation in aniridia-related keratopathy with a two-year follow-up period, an uncommon disease with precarious course. *Acta Ophthalmol.* **101**, 222-228 (2023).
13. Fuest, M., Jhanji, V. & Yam, G. H. Molecular and cellular mechanisms of corneal scarring and advances in therapy. *Int. J. Mol. Sci.* **24**, 7777 (2023).
17. De Arrigunaga, S. *et al.* Prospective, randomized, multicenter, double-masked, clinical trial of corneal cross-linking for Boston keratoprosthesis carrier tissue. *Am. J. Ophthalmol.* **249**, 39-48 (2023).
19. Moshirfar, M. *et al.* The historical development and an overview of contemporary keratoprotheses. *Surv. Ophthalmol.* **67**, 1175-1199 (2022).
20. Wang, L. *et al.* Injectable double-network hydrogel for corneal repair. *Chem. Eng. J.* **455**, 140698 (2023).
21. Li, Z., Wang, Q., Zhang, S. F., Huang, Y. F. & Wang, L. Q. Timing of glaucoma treatment in patients with MICOF: A retrospective clinical study. *Front. Med.* **9**, 986176 (2022).

Comment 2 from Reviewer #1:

The authors should provide more recent references regarding corneal transplantation

and keratoplasty.

Response: We sincerely appreciate your kind suggestion. We have added four recent references regarding corneal transplantation and keratoplasty. The reviewer may refer to the following content in the revised manuscript:

Page 2 and line 7: “Corneal transplantation of a donor allograft (i.e., keratoplasty) is the most common medical solution to this disease⁶⁻⁸ ... and 12.7 million patients are waiting for the process^{12,13}.”

Reference:

6. Bennett, V. G., Alberti, M., Quadrio, M. & Pralits, J. O. Optimization of patient positioning for improved healing after corneal transplantation. *J. Biomech.* **150**, 111510 (2023).
7. Lemaitre, D., Tourabaly, M., Borderie, V. & Dechartres, A. Long-term outcomes after lamellar endothelial keratoplasty compared with penetrating keratoplasty for corneal endothelial dysfunction: a systematic review. *Cornea* **42**, 917-928 (2023).
8. Viberg, A., Vicente, A., Samolov, B., Hjortdal, J. & Bystrom, B. Corneal transplantation in aniridia-related keratopathy with a two-year follow-up period, an uncommon disease with precarious course. *Acta Ophthalmol.* **101**, 222-228 (2023).
13. Fuest, M., Jhanji, V. & Yam, G. H. Molecular and cellular mechanisms of corneal scarring and advances in therapy. *Int. J. Mol. Sci.* **24**, 7777 (2023).

Comment 3 from Reviewer #1:

The cornea has not “feelings”. It does contain a large number of sensory nerve fibers that are responsible to transmit important sensations to the brain, such as touch, pressure, and temperature.

Response: We thank the reviewer for this comment. Cornea is the most densely innervated tissue in the human body^{R1,2}. The innervation of cornea is provided by the sensory fibers of the ophthalmic branch of the trigeminal nerve, and by the less-numerous sympathetic and parasympathetic nerve fibers^{R1,2}. Corneal sensation is critical in maintaining epithelial integrity and limbal stem cell function^{R3}. Reduced corneal sensation renders the corneal surface prone to injury and decreased reflex tearing^{R3}. Restoration of corneal sensation is one of the research hotspots in the medical field. It is of great enlightenment for corneal sensation reconstruction to realize the sensory function of the cornea using electronic devices. Thus, we fabricated an artificially-intelligent cornea that has tactile sensation in this work. The tactile sensation of the artificially-intelligent cornea makes it able to percept mechanical stimuli and realize corneal reflex that supply protection for the eyes. To make an appropriate expression, we have changed “an artificial cornea with “feelings”” to “an artificially-intelligent cornea with tactile sensation” in the revised manuscript. The reviewer may

refer to the following content in the revised manuscript:

We changed all “an artificial cornea with “feelings”” to “an artificially-intelligent cornea with tactile sensation”, such as:

The title: “An artificially-intelligent cornea with tactile sensation...”

The subheading: “An artificially-intelligent cornea with tactile sensation”

Page 18 and line 22: “Thus, we described the artificial corneal reflex arc as an artificially-intelligent cornea with tactile sensation.”

Reference

- R1. Belmonte, C., Acosta, M. C. & Gallar, J. Neural basis of sensation in intact and injured corneas. *Exp. Eye Res.* **78**, 513–525 (2004).
- R2. Labbe, A. *et al.* The relationship between subbasal nerve morphology and corneal sensation in ocular surface disease. *Invest. Ophthalmol. Vis. Sci.* **53**, 4926–4931 (2012).
- R3. Elbaz, U., Bains, R., Zuker, R. M., Borschel, G. H. & Ali, A. Restoration of corneal sensation with regional nerve transfers and nerve grafts: a new approach to a difficult problem. *JAMA Ophthalmol.* **132**, 1289–1295 (2014).

Comment 4 from Reviewer #1:

Figure 2A is cropped and does not show a representative sample of a biological synapse.

Response: Thank you very much for your meaningful comment. In response, we have changed Figure 2A. In the new Figure 2A, the biological synapse is shown more clearly, and its significant components are labeled, such as neurotransmitter, synaptic cleft, and receptor. The reviewer may refer to the following content in the revised manuscript:

Fig. 2 a, Schematic of a biological synapse.

Comment 5 from Reviewer #1:

Although several studies have shown the potential use of ZTO to create devices that could mimic the function of biological synapses, the term excitatory postsynaptic currents (EPSCs) should be restricted to recordings related to the release of neurotransmitters from presynaptic neurons. Perhaps a more suitable name could be:

“Artificial Excitatory Postsynaptic Currents (aEPSCs)”.

Response: We thank the reviewer for this comment. As suggested by the reviewer, we have changed the term “excitatory postsynaptic currents (EPSCs)” to “artificial excitatory postsynaptic currents (aEPSCs)” in the revised manuscript and the Supplementary Information.

The reviewer may refer to the following content in the revised manuscript and Supplementary Information:

In the revised manuscript:

Page 6 and line 10: “Artificial excitatory postsynaptic currents (aEPSCs)...”

We changed all “EPSC” to “aEPSC”

Figures: Figure 2c, Figures 3c, e, f, g, i, j, k, and m.

In the revised Supplementary Information:

We changed all “EPSC” to “aEPSC”

Supplementary Figures: Supplementary Figure 11b, c, Supplementary Figure 17, Supplementary Figure 18.

Comment 6 from Reviewer #1:

Information about the rise time of the artificial synapses should be provided.

Response: Thank you very much for your valuable comment. The rise time of the aEPSC of the artificial synapses is related to the settings in the test of synaptic devices. It is generally positively correlated with the duration of the presynaptic spike. As the duration of the presynaptic spike increased from 0.15 s to 0.55 s, the rise time of the aEPSC increased concurrently (Figure R1).

Figure R1. aEPSC of ZTO-3:7 artificial synapses triggered by presynaptic spikes (2V) with duration of 0.15, 0.25, 0.35, 0.45 or 0.55 s.

To clarify this point, we added a sentence in the revised manuscript, the reviewer may refer to the following content in the revised manuscript:

Page 6 and line 12: “The rise time of the aEPSC is related to the settings in the test of synaptic devices, and is generally positively correlated with the duration of the

presynaptic spike.”

Comment 7 from Reviewer #1:

The number of experiments performed regarding synaptic plasticity is unclear. More information should be provided. Furthermore, Figure 3 should show mean values and deviation measures.

Response: Thank you for your meaningful comment. Experiments performed regarding synaptic plasticity were repeated more than three times. According to your comment, we have added error bars in Figures 3d, 3f, 3h, and 3j to show the mean values and standard deviations.

We have added or modified several sentences and modified Figures 3d, 3f, 3h and 3j in the revised manuscript. The reviewer may refer to the following content in the revised manuscript:

Page 12 and line 16: “Experiments performed regarding synaptic plasticity were repeated more than three times. Corresponding statistics information was shown as mean values and standard deviations.”

Page 12 and line 21: “The mean maximum I_{PPF} (at $\Delta t = 50$ ms) were 133.0% for ZTO-7:3, 127.7% for ZTO-1:1, and 132.5% for ZTO-3:7ASs.”

Figures: Figures 3d, f, h and j.

Comment 8 from Reviewer #1:

Figure 4A is misleading. The optic nerve is misplaced.

Response: We apologize for the confusion. We have corrected the figure. To further facilitate audience comprehension, we have added a flowchart about the process of corneal reflex in Figure 4A. When the cornea is touched, sensory information is

transmitted to the spinal trigeminal nucleus through the trigeminal nerve (V). Then the information is transmitted to the facial nucleus from the spinal trigeminal nucleus through nerves that connect them. Finally, the efferent information is transmitted to the orbicularis oculi muscles through the facial nerve (VII) and induces eyelid closure. The reviewer may refer to the following content in the revised manuscript:

Fig.4 a, Schematic of corneal reflex examination (left), and the process of corneal reflex (right).

Comment 9 from Reviewer #1:

Although the cornea plays an important role in regulating the amount of light that enters the eye, it does not respond directly to changes in light levels; therefore, the conceptual design of the proposed artificial cornea is unclear. Additionally, the conceptual design and fabrication of the proposed approach in the biomedical field is missing.

Response: Thank you very much for your meaningful comment. We sincerely apologize for the confusion. The principal function of the cornea is to let light in to the eye and providing two-thirds of eye focusing power (function of light refraction)^{R4}. The cornea does not have a light-perception function, and can not respond directly to changes in light levels. The light perception function of the human eye relies on the retinal photoreceptors (rods and cones)^{R5,6}. The light regulation function (light adaptation) of the human eye relies on the light-adapted pupil size which is initiated by information transduction from the retinal photoreceptors^{R7-9}. The bright light triggers the sphincter pupillae to shrink, resulting in pupil contraction. Weak light triggers the dilator iridis to shrink, resulting in pupil dilation^{R10}.

In the revised manuscript, we have changed the statements about the light-perception and regulation functions of the artificially-intelligent cornea. Firstly, we fabricated an artificially-intelligent cornea with tactile sensation which can realize the corneal reflex like a human cornea under mechanical stimuli. Then, we endowed the artificially-intelligent cornea with a light perception function using light sensor-oscillator circuit, which can also interact with the environment due to that its transmittance can adapt to changes in the environment. The natural cornea does not have a light-perception function. However, the artificially-intelligent cornea can achieve this perception

function like the retina which relies on photoreceptors (rods and cones). Thus, we defined the light perception function of the artificially-intelligent cornea as “sensory expansion” (widening an existing sensory experience)^{R11}. The approaches of sensory substitution, augmentation, and expansion have been reported in numerous studies, which can help with visual reconstruction^{R12-17}. Thus, the artificially-intelligent cornea not only has tactile sensation like the native cornea, but also has light-perception function like the retina (i.e. the realizing of sensory expansion), and the capacity to interact with the environment.

To clarify the meaning on the paper, the subheading “Construction of artificial corneal reflex arc” was changed to “An artificially-intelligent cornea with tactile sensation”, and “Conceptual design and fabrication of an artificial cornea with ‘feelings’” was changed to “Sensory expansion and interactive functions of the artificially-intelligent cornea”. Correspondingly, the content has also been revised. The reviewer may refer to the following content in the revised manuscript:

In the revised manuscript:

Page 18 and line 19: “The artificial corneal reflex arc has protection, tactile perception, and light refraction functions like the native human cornea, and it can also mimic the contraction of the orbicularis oculi muscles. It is more intelligent than the native human cornea and the conventional artificial corneas that do not perceive tactile stimuli. Thus, we describe our artificial corneal reflex arc as an artificially-intelligent cornea with tactile sensation.”

Page 20 and line 1: “A natural cornea does not assume the functions of light perception and regulation of the changes in light levels. Light enters the eyes through the cornea, and is detected by the retinal photoreceptors (rods and cones), then the elongated photoreceptors convert the light signal to action potentials, which are transmitted to the brain through the optic nerve (Fig. 5a). To increase the smartness of our artificially-intelligent cornea with tactile sensation, we endowed it with light-perception ability by using a light sensor-oscillation circuit (Supplementary Fig. 21c, d) to replace the vibration sensor-oscillation circuit (Fig. 5c). The artificially-intelligent cornea realized sensory expansion, and obtained the ability to percept and respond to external light stimuli.”

Page 20 and line 10: “With the light illuminance increasing from 22 to 583 lx, the frequency of presynaptic spikes (the electric pulses output by the light sensor-oscillation circuit) increased from 0.5 Hz to 10 Hz (Fig. 5d). The statistical curve of signal output over time for each part of the artificially-intelligent cornea under light stimulation was also determined (Fig. 5e). The aEPSC of ZTO-3:7 AS triggered by different light illuminances induced the electrochromic actuator to remain light blue under 22 lx, and to change to dark blue under 583 lx. These results demonstrated that

the opening and closing state of human eyes were successfully imitated by an electrochromic actuator, which can realize the regulation of transmitted light and protection of human eyes (Fig. 5f, g). Furthermore, the transmittance of the electrochromic actuator can adapt to changes in the light intensity in the environment (Fig. 4f, Supplementary Movie 3). As light intensity increased, the frequency of presynaptic spikes increased, so the aEPSC increased and as a result, the transmittance of the electrochromic actuator decreased gradually. The light-adaption of transmittance demonstrated that the artificially-intelligent cornea can interact with the environment, to supply additional adaptive protection for the eyes in the changing environment. Therefore, the artificially-intelligent cornea simultaneously realized sensory expansion and interactive functions.”

Reference

- R4. Al-Aqaba, M. A., Dhillon, V. K., Mohammed, I., Said, D. G. & Dua, H. S. Corneal nerves in health and disease. *Prog. Retin. Eye Res.* **73**, 100762 (2019).
- R5. Liao, F. *et al.* Bioinspired in-sensor visual adaptation for accurate perception. *Nat. Electron.* **5**, 84–91 (2022).
- R6. Miller, R. E. & Tredici, T. J. Night vision manual for the flight surgeon. (ARMSTRONG LAB BROOKS AFB TX, 1992).
- R7. Laeng, B. & Sulutvedt, U. The eye pupil adjusts to imaginary light. *Psychol. Sci.* **25**, 188–197 (2014).
- R8. Nicholls, J. G., Martin, A. R., Wallace, B. G. & Fuchs, P. A. *From neuron to brain*. Vol. 271 (Sinauer Associates Sunderland, MA, 2001).
- R9. Markwell, E. L., Feigl, B. & Zele, A. J. Intrinsically photosensitive melanopsin retinal ganglion cell contributions to the pupillary light reflex and circadian rhythm. *Clin. Exp. Optom.* **93**, 137–149 (2010).
- R10. Gong, J. *et al.* An artificial visual nerve for mimicking pupil reflex. *Matter* **5**, 1578–1589 (2022).
- R11. Eagleman, D. M. & Perrotta, M. V. The future of sensory substitution, addition, and expansion via haptic devices. *Front. Hum. Neurosci.* **16**, 1055546 (2022).
- R12. Nau, A., Bach, M. & Fisher, C. Clinical tests of ultra-low vision used to evaluate rudimentary visual perceptions enabled by the brainport vision device. *Transl. Vis. Sci. Technol.* **2**, 1 (2013).
- R13. Maidenbaum, S., Abboud, S. & Amedi, A. Sensory substitution: closing the gap between basic research and widespread practical visual rehabilitation. *Neurosci. Biobehav. Rev.* **41**, 3–15 (2014).
- R14. Zhang, Y., Chen, Y., Zhao, B., Hu, Y. & Li, Z. in *2018 IEEE Micro. Electro. Mechanical Systems (MEMS)*. 415–418.
- R15. Longin, L. & Deroy, O. Augmenting perception: How artificial intelligence transforms sensory substitution. *Conscious. Cogn.* **99**, 103280 (2022).
- R16. Shull, P. B. & Damian, D. D. Haptic wearables as sensory replacement, sensory augmentation and trainer – a review. *J. Neuroeng. Rehabil.* **12**, 59 (2015).

R17. Karcher, S. M., Fenzlaff, S., Hartmann, D., Nagel, S. K. & König, P. Sensory augmentation for the blind. *Front. Hum. Neurosci.* 6, 37 (2012).

Comment 10 from Reviewer #1:

As stated before, the cornea does not have any photoreceptor cells that detect light levels. Therefore, the schematic of the corneal reflex triggered by bright light (Fig 5) is misleading.

Response: We sincerely appreciate your meaningful comment and apologize for the confusion. We have revised Figure 5a; as it shows now, the light enters the eyes through the cornea, then photoreceptor cells on the retina detect light level, then the signals were transported to the brain by the optic nerve. The reviewer may refer to the following content in the revised manuscript:

Fig. 5a Schematic of the light perception in the human eyes.

Comment 11 from Reviewer #1:

The conclusions are not supported by the results.

Response: We sincerely appreciate your comment. We have removed or changed some claims, for instance, “It may help restore the vision in individuals who have corneal blindness”. We have rewritten the Abstract, Introduction, and Conclusions according to the experiments and results. We believe that the revised conclusions are supported by the results in the revised manuscript. The reviewer may refer to the following content in the revised manuscript:

Abstract: “We demonstrate an **artificially-intelligent** cornea that can assume the functions of the native human cornea such as protection, **tactile perception**, and **light refraction**, and **possesses sensory expansion and interactive functions**. This work suggests new strategies for the tuning of synaptic plasticity and the design of visual neuroprosthetics, and **has important implications for the development of neuromorphic electronics and for visual restoration**.”

Introduction: “This paper describes an **artificially-intelligent** cornea that has the functions of protection, tactile perception and light refraction **like the native human cornea**, and also has **sensory expansion (widening an existing sensory experience) and interactive functions**. The **touch sensitivity** of the This work presents a new

resource that may be used in development of powerful neuromorphic electronics and visual neuroprosthetics.”

Page 20 and line 27: “As a proof-of-concept, a robot was equipped with the **artificially-intelligent** cornea (Fig. 5b), which was light blue with high transmittance under weak light, and changed to dark blue with low transmittance under bright light (Fig. 5h). Notably, the **artificially-intelligent** cornea is logically compatible with the human nervous system, replicates protection, tactile perception, and light refraction functions of the native human cornea, and expands light-perception and interactive functions. In the future, the development tendency of the novel **artificially-intelligent** cornea will be oriented towards biocompatibility, stability, miniaturization, and high integration. After development and optimization, the **artificially-intelligent** cornea may be compatible with transplantation into individuals who have corneal blindness and are waiting for medical interventions, thereby alleviating the shortage of donor corneas. We believe that the mature **artificially-intelligent** cornea has potential applications in neuroprosthetics and visual rehabilitation.”

Conclusions: “In summary, we have developed an **artificially-intelligent** cornea that replicates the functions of the native human cornea such as protection, tactile perception, and light refraction, and expands light perception and interactive functions. To realize these functions, an artificial corneal reflex arc that can implement mechanical and light information coding, information processing, and the regulation of transmitted light was constructed by integrating sensor-oscillation circuits, ZTO ASs, and electrochromic devices. In the future, the biocompatibility, stability, size, and integration degree of the **artificially-intelligent** cornea must be optimized. The mature **artificially-intelligent** cornea may have applications in neuroprosthetics and visual restoration.”

Comment 12 from Reviewer #1:

Discussion should include possible limitations of the study.

Response: Thank you very much for your valuable comment. In this paper, we have developed an **artificially-intelligent** cornea that replicates the functions of the native human cornea such as protection, tactile perception, and light refraction, and expands light perception and interactive functions. However, many aspects of the **artificially-intelligent** cornea still require improvement and optimization; examples include biocompatibility, stability, miniaturization, and integration degree. To follow your comment, we have discussed the possible limitations of this study in the revised manuscript. The reviewer may refer to the following content in the revised manuscript: **Page 21 and line 3:** “In the future, the development tendency of the novel **artificially-intelligent** cornea will be oriented towards biocompatibility, stability, miniaturization,

and high integration.”

Page 21 and line 23: “In the future, the biocompatibility, stability, size, and integration degree of the artificially-intelligent cornea must be optimized.”

Comment 13 from Reviewer #1:

There are several typos, such as "A stimulation voltages were increased... (pag#15). English should be reviewed.

Response: We apologize for our carelessness. We have reviewed English and corrected the typos in the revised manuscript. Reviewer may refer to the following content in the revised manuscript:

Page 6 and line 14: “Upon the arrival ... or become embedded ... an increase... attributed to ...”

Page 8 and line 19: “... structures lead to an increase ...”

Page 9 and line 14: “The increase in the number ... μ_n [cm²/(V·s)] was calculated ...”

Page 10 and line 13: “These phenomena are consistent with...”

Page 12 and line 19: “... ZTO-7:3, ZTO-1:1, and ZTO-3:7 ASs ... of the increase in...”

Page 13 and line 26: “... and “dash (-)” ... as “dash (-)” ... The input letters ... the input letters ...”

Page 14 and line 2: “... the LTP of ZTO-3:7 AS was exploited to ... An aEPSC=14.5 μ A ... salivation; this result indicates that ...”

Page 16 and line 21: “Physicians ... by examining the corneal reflex.”

Page 16 and line 23: “a ZTO-3:7 AS, an amplifier circuit, and a pair of electrochromic devices”

Page 16 and line 29: “As the processing core, the ZTO-3:7 AS was selected due to its excellent synaptic activity.”

Page 17 and line 5: “The sandwich-type ... an electrochromic layer that uses PEDOT: PSS, and a gel electrolyte layer ...”

Page 17 and line 10: “As stimulation voltages were increased from 0 V to 3 V, the color of the electrochromic device...and consequently its transmittance ...”

Reviewer #2:

In the paper authored by Qu et al., a novel device and system design is presented to emulate the reflex reactions of the cornea in closing eyelids. The system comprises four modules: a sensor-oscillator circuit, a ZTO-nanofiber-based synaptic device, an amplified circuit, and an electrochromic device. The application of the electrochromic

device to mimic eyelid closing functionality is particularly intriguing. The ZTO nanofiber provides excellent transparency and tunable performance for the synaptic device, allowing for leaky integration of the pulsed sensor output. This work is anticipated to inspire further advancements in the application of neuromorphic computing concepts to artificial eye system development. To further consider this work for publication, the authors should address the following comments:

Response: We sincerely appreciate your meaningful comments. In response to the reviewer's comments and suggestions, we have performed additional experiments to improve the quality of our work and have added three brief video files to facilitate audience comprehension. The point-by-point responses to the reviewer's comments are presented below.

Comment 1 from Reviewer #2:

Comprehensive performance characterization results for both touch and optical sensor-oscillator circuits are required, particularly with measurements of circuit output under gradually varied pressure or light intensity.

Response: We appreciate your insightful comments on our research. We provide detailed information about the characterization of the electrical performance of vibration and light sensor-oscillator circuits (Figure R2). For the vibration test, mechanical stimulation was simulated using a point touch; for the light-stimulation test, the illuminance was distributed from 22 to 583 lx.

Figure R2. Characterization of the electrical performance of (a) vibration and (b) light sensor-oscillator circuits.

We modified the original Fig. 4i and Fig. 5e (new Supplementary Figure 24 and new Fig. 5e), and revised the corresponding content in the revised manuscript and Supplementary Information. The reviewer may refer to the following content in the revised manuscript and Supplementary Information:

In the revised manuscript:

Page 17 and line 20: “The statistical curve of signal output over time for each part of the artificial corneal reflex arc under mechanical stimulation was determined

(Supplementary Fig. 24). When a finger or a foreign object touches the vibration sensor, it acts as a normally closed device that conducts at this time, making the power-supply side of the oscillator circuit turn on and output a high-frequency spikes pulse. The output spike signals are used as the presynaptic stimulus to trigger the postsynaptic current of the artificial synapse, then an amplifier circuit outputs a voltage signal, which is used as the modulation signal of the electrochromic unit. Gradual changes of electrochromic actuators under the real-time touch can be seen in Supplementary Movie 1.”

Page 20 and line 12: “The statistical curve of signal output over time for each part of the artificially-intelligent cornea under light stimulation was also determined (Fig. 5e).”
Figures: Fig 5e.

Fig. 5e. Statistical curve of signal output over time for each part of the artificially-intelligent cornea under light stimulation.

In the revised Supplementary Information:

Supplementary Figure 24. Statistical curve of signal output over time for each part of the artificial corneal reflex arc under mechanical stimulation.

Comment 2 from Reviewer #2:

The paper should include temporal response information for the electrochromic device and discuss the extent to which this response speed can replicate the eyelid closure process.

Response: We sincerely appreciate your valuable comment. According to your comment, we have performed experiments to quantify the temporal response information of the electrochromic device. For the test, we used a Silicon photodetector to collect the optical signal that was transmitted through the electrochromic device (Supplementary Fig. 23a). The transmittance change of the electrochromic device was quantified by the change of photocurrent collected from the photodetector. The response time was extracted from time-dependent photocurrent as the electrochromic device changed color. The device took ~ 4.3 s to achieve a 90 % change in transmittance (Supplementary Fig. 23b). In addition, the electrochromic device can respond to the voltage (3 V) with different duration (50, 100, and 150 ms) (Supplementary Fig. 23c). As is reported, the eyelid closure needs 100 \sim 150 ms^{R18}. Thus, the response speed of the electrochromic device can replicate the eyelid-closure process.

We have added several sentences, references, and one figure in the revised manuscript and the Supplementary Information. The reviewer may refer to the following content in the revised manuscript and the Supplementary Information:

In the revised manuscript:

Page 17 and line 13: “ The electrochromic device took ~ 4.3 s to achieve a 90 % change in transmittance under 3 V, and it can respond to the voltage with a duration of ~ 50 ms (Supplementary Fig. 23), which is less than the duration of eyelid closure (100 \sim 150

ms)⁶⁴. Thus, the response speed of the device can replicate the eyelid-closure process.”

Reference:

64. Burr, D. Vision: in the blink of an eye. *Curr. Biol.* **15**, R554–556 (2005).

In the revised Supplementary Information:

Supplementary Figure 23. Measurement of the temporal response information of the electrochromic device. **a** Schematic of the configuration in the test. **b** Time-dependent photocurrent during coloring of the electrochromic device. **c** Response of the electrochromic device under the applied voltages with different duration (50, 100, and 150 ms). The transmittance change of the electrochromic device was displayed by the change of photocurrent collected from the photodetector. The response time was extracted from time-dependent photocurrent during color change of the electrochromic device.

Reference

R18. Burr, D. Vision: in the blink of an eye. *Curr. Biol.* **15**, R554–556 (2005).

Comment 3 from Reviewer #2:

To assess the operation of the demonstrated artificial corneal system, signal measurements at each stage of the signal processing path (e.g., after the synaptic device and amplification) are necessary.

Response: We appreciate your insightful comments on our research. The schematic diagram and corresponding statistical curves of the signal output of each part of the artificial cornea reflex arc are shown in Figure R3.

Figure R3. (a) Schematic diagram of the signal output of each part of the artificial corneal reflex arc. The statistical curve of signal output over time for each part of the artificial corneal reflex arc under (b) mechanical and (c) light stimulation.

To follow your comments, we have added relevant results in the revised manuscript and Supplementary Information. We modified the original Fig. 4i and Fig. 5e (new Supplementary Figure 24 and new Fig. 5e), and added the corresponding description in the revised manuscript and Supplementary Information:

In the revised manuscript:

Page 17 and line 20: “The statistical curve of signal output over time for each part of the artificial corneal reflex arc under mechanical stimulation was determined (Supplementary Fig. 24). When a finger or a foreign object touches the vibration sensor, it acts as a normally closed device that conducts at this time, making the power-supply side of the oscillator circuit turn on and output a high-frequency spikes pulse. The output spike signals are used as the presynaptic stimulus to trigger the postsynaptic current of the artificial synapse, then an amplifier circuit outputs a voltage signal, which is used as the modulation signal of the electrochromic unit. Gradual changes of electrochromic actuators under the real-time touch can be seen in Supplementary Movie 1.”

Page 20 and line 12: “The statistical curve of signal output over time for each part of the artificially-intelligent cornea under light stimulation was also determined (Fig. 5e).”
 Figures: Fig 5e.

Fig. 5e. Statistical curve of signal output over time for each part of the artificially-intelligent cornea under light stimulation.

In the revised Supplementary Information:

Supplementary Figure 24. Statistical curve of signal output over time for each part of the artificial corneal reflex arc under mechanical stimulation.

Comment 4 from Reviewer #2:

The authors should elaborate on how this system can achieve the four distinct reflex types mentioned in the paper and depicted in Fig. 4k.

Response: We sincerely appreciate this comment. Dysfunction of the corneal reflex can be caused by damage to the central or peripheral nervous system in the pathway. When the cornea reflex arc without dysfunction, the corneal reflex can realize. Afferent pathway damage or central dysfunctions cause deficits in corneal reflex bilaterally;

contralateral and ipsilateral efferent damages correspond to ipsilateral and contralateral reflex, respectively. In our system, when the stimulus was applied to the right eye, all parts operated normally for the bilateral corneal reflex. The disconnection of the afferent pathway or synaptic device causes deficits in the corneal reflex bilaterally. The disconnection of the left efferent pathway and the right efferent pathway between the synaptic device and electrochromic actuators results in the dysfunction of the contralateral and ipsilateral reflex, respectively.

The original Fig. 4k only shows the picture of the electrochromic actuators, it does not show the full picture of the system. Thus, to make it clear, we added several sentences in the revised manuscript to elaborate on how this system can achieve the four distinct reflex types and added Supplementary Figure 25 in the revised Supplementary Information to depict. The reviewer may refer to the following content in the revised manuscript and the Supplementary Information:

In the revised manuscript:

Page 18 and line 5: “(Fig. 4i, Supplementary Fig. 25, Supplementary Movie 2), all parts of the system operated normally, and”

Page 18 and line 9: “the afferent pathway or synaptic device in the system was disconnected, and”

Page 18 and line 12: “When the left efferent pathway between the synaptic device and electrochromic actuators was disconnected, the right electrochromic actuator changed to dark blue (Fig. 4i III). In contrast, when the right efferent pathway between the synaptic device and electrochromic actuators was disconnected, the left electrochromic actuator changed to dark blue (Fig. 4i IV).”

In the revised Supplementary Information:

Supplementary Figure 25. Schematic of the artificial corneal reflex arc evoked by mechanical stimuli.

Comment 5 from Reviewer #2:

In addition to ZTO transparency, it is essential to measure the transparency of the entire synaptic device.

Response: Thank you very much for your valuable suggestion. According to your suggestion, we have measured the transparency of the entire synaptic device prepared on a glass substrate. As shown in Supplementary Figure 3a, the ITO electrodes, ZTO-3:7 fibers, and Ion gel were used as drain and source electrodes, channel, and gate dielectrics, respectively. The transparency of the entire synaptic device was $\sim 78.91\%$ at the wavelength of 550 nm (Supplementary Fig. 3a). The electrical performance of the transparent synaptic device was also measured. Supplementary Figures 3b, c, and d show the aEPSC of the transparent synaptic device triggered by a single spike (3.5 V, 50 ms), a pair of spikes (3.5 V, 50 ms), and spikes with amplitudes 0.5, 1, 1.5, 2, 2.5, 3, 3.5, 4, 4.5, 5, and 5.5 V.

We added one sentence to the revised manuscript and one figure to the revised Supplementary Information. The reviewer may refer to the following content in the revised manuscript and the Supplementary Information:

In the revised manuscript:

Page 8 and line 3: “The transparency of the entire synaptic device prepared on a glass substrate was $\sim 78.91\%$ at $\lambda = 550$ nm (Supplementary Fig. 3).”

In the revised Supplementary Information:

Supplementary Figure 3. Transparency and synaptic plasticity of transparent synaptic device. a Structure and transparency of the entire synaptic device prepared on

a glass substrate. **b** aEPSC of the transparent synaptic device triggered by a single spike (3.5 V, 50 ms) at $V_d = 2$ V. **c** aEPSC of the transparent synaptic device triggered by a pair of spikes (3.5 V, 50 ms) at Δt (time intervals between two spikes) = 100 ms. **d** aEPSC of the transparent synaptic device triggered by spikes with amplitudes 0.5, 1, 1.5, 2, 2.5, 3, 3.5, 4, 4.5, 5, and 5.5 V.

Note: The transparent synaptic device is composed of a glass substrate, patterned ITO electrodes, ZTO-3:7 fibers, and ion gel. The fabrication process of the transparent synaptic device is similar to the ZTO-3:7 artificial synapse prepared on the SiO₂/Si substrate except for the source and drain electrodes. The source and drain electrodes of the transparent synaptic device were patterned ITO obtained by sputtering. The width of ITO electrodes is 3 mm, and the length between source and drain electrodes is 4 mm.

Comment 6 from Reviewer #2:

To facilitate audience comprehension, the authors should consider including videos that demonstrate the real-time touch/light-induced switching of the electrochromic devices in conjunction with figures 4k, 5g, and 5h.

Response: We sincerely appreciate your suggestion. By following your suggestion, we have added three brief video files that demonstrate the real-time touch/light-induced switching of the electrochromic devices (Supplementary Movies 1-3). The reviewer may refer to the following content in the revised manuscript and the Supplementary Movies:

In the revised manuscript:

Page 17 and line 28: “Gradual changes of electrochromic actuators under the real-time touch can be seen in Supplementary Movie 1.”

Page 18 and line 5: “(Fig. 4i, Supplementary Fig. 25, Supplementary Movie 2)”

Page 20 and line 20: “(Fig. 4f, Supplementary Movie 3)”

In the Supplementary Movie:

Supplementary Movie 1: Gradual changes of electrochromic actuators under the real-time touch;

Supplementary Movie 2: States of electrochromic actuators under bilateral reflex, ipsilateral reflex and contralateral reflex;

Supplementary Movie 3: Gradual changes of electrochromic actuators under different light input intensities.

Comment 7 from Reviewer #2:

As the ultimate objective is to integrate such systems with patients or even soft robots, the importance of soft and stretchable properties cannot be understated. The authors are

encouraged to address this aspect in the paper, referring to recent works on developing stretchable neuromorphic devices for human-interfaced applications.

Response: Thank you for your meaningful suggestion. We agree that the importance of soft and stretchable properties cannot be understated. Recently, the flexible and stretchable neuromorphic devices for bio or human-interfaced applications were widely reported, which open the door for a future direction toward neurorehabilitation and more powerful biomimetics^{R19-22}.

Using our material and technology, a flexible artificial synapse was prepared, and its soft properties (Supplementary Fig. 20) were investigated, which has potential applications in flexible electronics and human-interface. The fabrication process of the flexible artificial synapse is as follows:

A polyimide (PI) substrate was ultrasonically cleaned in deionized water, isopropanol, and anhydrous ethanol for 30 min, sequentially. Then digitally-aligned ZTO fibers (molar ratio of Zn: Sn = 3:7) were printed on the substrate surface to produce the channel. The printed ZTO fibers were annealed in a muffle furnace at 350 °C for 2 h. Gold source and drain electrodes (80 nm) were then thermally deposited through a shadow mask. Finally, the prepared PVDF-HFP/[EMIM-TFSI] ion gel was transferred onto the ZTO fibers channel area.

The flexible characteristics of the device have been investigated by performing experiments on bending stability. The device was repeatedly bent using a desktop bending test machine, and the bending times were 0, 500, 1000, 1500, and 2000 (Supplementary Fig. 20a). For the static bending test, the device was attached to the surface of a plastic cylinder with a radius of 1 cm to conduct (Supplementary Fig. 20b). The aEPSC of the flexible ZTO-3:7 artificial synapse under different bending states (0, 500, 1000, 1500, and 2000 bending times, bent at 1 cm radius) shows a little degradation (Supplementary Fig. 20c). The current retention was defined as the ratio of aEPSC peak with/without bending. The current retention was > 93% (< 7 % loss) after 2000 times bending and was > 98% (< 2 % loss) after static bending to a radius of 1 cm (Supplementary Fig. 20d). These results demonstrated the excellent bending resistance of the flexible artificial synapse. Paired-pulse facilitation and spike-number-dependent plasticity of the flexible artificial synapse were also investigated (Supplementary Fig. 20e, f).

We added several sentences and references in the revised manuscript and Supplementary Information. The reviewer may refer to the following content in the revised manuscript and Supplementary Information:

In the revised manuscript:

Page 14 and line 12: “The flexible and stretchable properties of artificial synapses are of great significance for wearable and human-interface applications⁵³⁻⁵⁶. Thus, we fabricated a flexible ZTO-3:7 AS on a polyimide (PI) substrate and investigated its

flexible characteristics. The synaptic properties of the device show excellent bending resistance (< 7% variation after 2000 times bending), so it has potential applications in flexible electronics (Supplementary Fig. 20).”

4 references:

53. Lee, Y. *et al.* A low-power stretchable neuromorphic nerve with proprioceptive feedback. *Nat. Biomed. Eng.* **7**, 511–519 (2023).
54. Shim, H. *et al.* Artificial neuromorphic cognitive skins based on distributed biaxially stretchable elastomeric synaptic transistors. *Proc. Natl. Acad. Sci. USA* **119**, e2204852119 (2022).
55. Wang, W. *et al.* Neuromorphic sensorimotor loop embodied by monolithically integrated, low-voltage, soft e-skin. *Science* **380**, 735–742 (2023).
56. Jiang, C. *et al.* Mammalian-brain-inspired neuromorphic motion-cognition nerve achieves cross-modal perceptual enhancement. *Nat. Commun.* **14**, 1344 (2023).

In the revised Supplementary Information:

Supplementary Figure 20. Synaptic properties of the flexible ZTO-3:7 artificial synapse. **a** Digital image of the cyclic bending test of the device (bending 0, 500, 1000, 1500, and 2000 times). **b** Digital image of static bending test of the device (bending radius = 1 cm). **c** aEPSC of the flexible ZTO-3:7 artificial synapse under different bending states (0, 500, 1000, 1500, and 2000 bending times, bent at 1-cm radius). **d** Current retention (the ratio of aEPSC peak with/without bending) under different bending states. **e** aEPSC of the flexible ZTO-3:7 artificial synapse triggered by a pair of spikes (3.5 V, 50 ms) at Δt (time intervals between two spikes) = 50 ms. **f** aEPSC of the flexible ZTO-3:7 artificial synapse triggered by consecutive spikes (3.5 V, 50 ms) with spike numbers 2, 3, 4, 5, 6, 7, 8, 9, and 10.

Note: The fabrication process of the flexible artificial synapse is as follows:

A polyimide (PI) substrate was ultrasonically cleaned in deionized water, isopropanol,

and anhydrous ethanol for 30 min, sequentially. Then digitally-aligned ZTO fibers (molar ratio of Zn:Sn = 3:7) were printed on the substrate surface to produce the channel. The printed ZTO fibers were annealed in a muffle furnace at 350 °C for 2 h. Gold source and drain electrodes (80 nm) were then thermally deposited through a shadow mask. Finally, the prepared PVDF-HFP/[EMIM-TFSI] ion gel was transferred onto the ZTO-fibers channel area.

Reference

- R19. Lee, Y. *et al.* A low-power stretchable neuromorphic nerve with proprioceptive feedback. *Nat. Biomed. Eng.* **7**, 511–519 (2023).
- R20. Shim, H. *et al.* Artificial neuromorphic cognitive skins based on distributed biaxially stretchable elastomeric synaptic transistors. *Proc. Natl. Acad. Sci. USA* **119**, e2204852119 (2022).
- R21. Wang, W. *et al.* Neuromorphic sensorimotor loop embodied by monolithically integrated, low-voltage, soft e-skin. *Science* **380**, 735–742 (2023).
- R22. Jiang, C. *et al.* Mammalian-brain-inspired neuromorphic motion-cognition nerve achieves cross-modal perceptual enhancement. *Nat. Commun.* **14**, 1344 (2023).

Reviewer #3:

This research has brought a novel concept in design of artificial cornea with the so-called feeling sensibility. The idea of tailoring an electro-chromatic system into an integrated system for development of a light-sensitive nociceptive system is interesting. However, the present style of manuscript needs considerable improvement to be considered as a high-quality research work that fulfills the expected standards. Generally, you should provide enough details on the methods and working mechanism of your system. The work is not reproducible in the present style. Authors are invited to go through the following recommendations for improving the quality of their research work via major revision:

Response: The detailed evaluation of our work from this reviewer is highly appreciated. In response to the detailed reviewer comments, we have performed additional experiments to improve the quality of this work, have provided more details to enable the reproducibility of our approach, and have revised the title and main manuscript to satisfy the quality of a well-structured manuscript. Please find below our responses (in blue) to each of your specific comments (in black). The revisions to the original manuscript and Supplementary Information are highlighted in red.

Comment 1 from Reviewer #3:

The title of manuscript is not informative. It is necessary to include more information

in title to reflect the mechanism of performance of this system.

Response: We agree with the reviewer's comment on the title. We have modified it to make it informative. The artificially-intelligent cornea that can realize a corneal reflex depends on three core components: sensor-oscillation circuits as the receptors that transform external stimuli to impulse spikes, ZTO artificial synapse as the processing core that transfers and integrates information, and electrochromic device as the actuator (eyelid) that respond to the postsynaptic current. The tactile sensation of the system was realized by the vibration sensor-oscillation circuit. In addition, the light sensor-oscillation circuit endows it with a light perception function. Light perception is not the function of the native human cornea, so artificially-intelligent cornea realized sensory expansion, which can further interact with the environment and supply protection for the eyes in the changing environment. Thus, the title has been changed to "An artificially-intelligent cornea with tactile sensation enables sensory expansion and interaction". The reviewer may refer to the following content in the revised manuscript and Supplementary Information:

Title: "An artificially-intelligent cornea with tactile sensation enables sensory expansion and interaction"

Comment 2 from Reviewer #3:

In introduction section" It may help restore the vision in individuals who have corneal blindness" This a powerful claim that is hard to be grasp from the findings of manuscript. You may use the similar expressions in conclusion, but this is not practically proved.

Response: We sincerely appreciate your comment. By following this comment, we have removed this expression in the introduction section of the revised manuscript. "Helping to restore the vision in individuals who have corneal blindness" is the potential application of the artificially-intelligent cornea. However, to realize this ultimate objective, many aspects remain to be improved and optimized, such as biocompatibility, stability, miniaturization, and integration degree. Thus, we use similar expressions in the revised manuscript to prospect the future application of the artificially-intelligent cornea. The reviewer may refer to the following content in the revised manuscript:

Page 21 and line 3: "In the future, the development tendency of the novel artificially-intelligent cornea will be oriented towards biocompatibility, stability, miniaturization, and high integration. After development and optimization, the artificially-intelligent cornea may be compatible with transplantation into individuals who have corneal blindness and are waiting for medical interventions, thereby alleviating the shortage of donor corneas."

Page 21 and line 23: “In the future, the biocompatibility, stability, size, and integration degree of the artificially-intelligent cornea must be optimized. The mature artificially-intelligent cornea may have applications in neuroprosthetics and visual restoration.”

Comment 3 from Reviewer #3:

The introduction section needs considerable alteration and improvement. The present style is not informative and cannot satisfy the quality of a well-structured manuscript. Authors are invited to alter their approach toward the presentation of proposed challenges, the importance of their innovative idea, and their strategy to tackle the challenge. They may also go through different research papers that proposed different approaches and mechanisms for design of artificial cornea. Artificial cornea have developed previously, please explain the previous achievements and then elaborate your approach and solving strategy. Will the work be of significance to the field and related fields? How does it compare to the previous published works?

Response: Thank you very much for your valuable comment. By following this comment, we have rewritten the introduction section and added several references. We believe the introduction section in the revised manuscript is informative and can satisfy the quality of a well-structured manuscript. The reviewer may refer to the following content in the revised manuscript:

Introduction:

“Located at the front of the eye, a cornea is a transparent structure that provides focusing power and shields the iris and lens from foreign substances¹. It is the most densely-innervated part of the body, and is therefore sensitive to foreign contaminants; a touch of the cornea causes an involuntary reflex to close the eyelid (the corneal reflex)². However, corneal diseases can result in blindness, and more than 10 million individuals worldwide suffer from bilateral corneal blindness³⁻⁵. Corneal transplantation of a donor allograft (i.e., keratoplasty) is the most common medical solution to this disease⁶⁻⁸, but due to scarcity of donated corneas, it can be used on only 1 in 70 patients⁹⁻¹¹, and 12.7 million patients are waiting for the process^{12,13}.

As a solution to this scarcity, artificial corneal substitutes that comply with the requirements for optical and transparency have been developed¹⁴. The Boston keratoprosthesis (Kpro) is the most-commonly implanted artificial corneal substitute worldwide; it is a collar-button-shaped device that is mainly made of polymethyl methacrylate (PMMA)¹⁵⁻¹⁷. Aurolab Kpro shares the design with Boston Kpro and is considered an alternative to the latter when affordability is a limiting factor¹⁸. MICO F Kpro contains a central ring with a threaded PMMA optic that is supported by a titanium frame, and the surgical invasiveness is mild¹⁹⁻²¹. As a unique approach to the artificial cornea, the osteo-odonto-keratoprosthesis is a biological Kpro that uses the lamina derived from an autologous mono-radicular tooth as a frame for PMMA optical

cylinder^{22,23}. All of these types of artificial cornea have been applied clinically to improve vision in cases of severe corneal opacification, and the effects of visual restoration, long-term safety, practicality, and cost are acceptable to the patients.

Although existing artificial corneas can assume partial functions of the native human cornea, such as protection and light refraction, they do not reconstruct the tactile sensation and therefore do not realize the corneal reflex. Thus, trying to develop a ‘smarter’ artificial cornea with tactile sensation and even possessing sensory expansion and interactive functions is of great significance for restoring vision in corneal blindness. Moreover, these functions must be realized by ‘invisible’ built-in electronics²⁴⁻²⁸, to achieve the high transparency and low haze of a natural cornea.

This paper describes an **artificially-intelligent** cornea that has the functions of protection, tactile perception and light refraction **like the native human cornea, and also has sensory expansion (widening an existing sensory experience) and interactive functions**. The **touch sensitivity** of the cornea was reconstructed by a reflex arc composed of sensor-oscillation circuits, zinc tin oxide (ZTO) artificial synapses (ASs) and electrochromic devices, which implement mechanical and light information coding, information processing and the regulation of transmitted light. Heavy-metal-free ZTO fabric patterns were fabricated with long, continuous morphology and digital alignment, and used as active channels due to their non-toxicity, low-cost, high transmittance (> 99.89%), and low haze (< 0.36%) to ensure the optical properties of the **artificial synapse**. The crystal phase of the ZTO fibers was highly tuned, and therefore ensured tunable synaptic plasticity, and incidental applications to encrypted communication and associative learning. **This work presents a new resource that may be used in development of powerful neuromorphic electronics and visual neuroprosthetics.”**

Reference:

6. Bennett, V. G., Alberti, M., Quadrio, M. & Pralits, J. O. Optimization of patient positioning for improved healing after corneal transplantation. *J. Biomech.* **150**, 111510 (2023).
7. Lemaitre, D., Tourabaly, M., Borderie, V. & Dechartres, A. Long-term outcomes after lamellar endothelial keratoplasty compared with penetrating keratoplasty for corneal endothelial dysfunction: a systematic review. *Cornea* **42**, 917-928 (2023).
8. Viberg, A., Vicente, A., Samolov, B., Hjortdal, J. & Bystrom, B. Corneal transplantation in aniridia-related keratopathy with a two-year follow-up period, an uncommon disease with precarious course. *Acta Ophthalmol.* **101**, 222-228 (2023).
13. Fuest, M., Jhanji, V. & Yam, G. H. Molecular and cellular mechanisms of corneal scarring and advances in therapy. *Int. J. Mol. Sci.* **24**, 7777 (2023).
17. De Arrigunaga, S. *et al.* Prospective, randomized, multicenter, double-masked, clinical trial of corneal cross-linking for Boston keratoprosthesis carrier tissue. *Am. J. Ophthalmol.* **249**, 39-48 (2023).

19. Moshirfar, M. *et al.* The historical development and an overview of contemporary keratoprostheses. *Surv. Ophthalmol.* **67**, 1175–1199 (2022).
20. Wang, L. *et al.* Injectable double-network hydrogel for corneal repair. *Chem. Eng. J.* **455**, 140698 (2023).
21. Li, Z., Wang, Q., Zhang, S. F., Huang, Y. F. & Wang, L. Q. Timing of glaucoma treatment in patients with MICOF: A retrospective clinical study. *Front. Med.* **9**, 986176 (2022).

Comment 4 from Reviewer #3:

Author are recommended to label 3 different components of their system in Figure 1, i.e. the sensor-oscillation circuit, processing core, actuator.

Response: Thank you for your kind suggestion. We have labeled 3 different components of the system in Figure 1, i.e. the sensor-oscillation circuits, artificial synapse, and electrochromic actuator. The reviewer may refer to the following content in the revised manuscript:

Fig. 1

Comment 5 from Reviewer #3:

It is necessary to provide the statistics information about the number, and coverage area of surface with ZTO fibers. We should know how many ZTO fibers are deposited, and then determine what the contact area is between the ionic gel and the ZTO fibers.

Response: We sincerely appreciate this valuable comment. We supplied the digital image and optical microscopy images of ZTO artificial synapses (Supplementary Fig.1). Each ZTO artificial synapse includes 19 ZTO fibers that contact the gold interdigital electrodes (Supplementary Fig.1a, 1b). The distance between two adjacent interdigital electrodes is $\sim 50 \mu\text{m}$ (Supplementary Fig.1c). The fibers between two adjacent

interdigital electrodes are exposed and covered with ion gel, and the rest are covered with gold electrodes (Supplementary Fig.1d). Each ZTO fiber exposes 11 segments to the ion gel, so the total length of one fiber covered with ion gel is $\sim 550 \mu\text{m}$. For a ZTO artificial synapse, the total length of ZTO fibers covered with ion gel is $\sim 10450 \mu\text{m}$. According to the SEM image of Figure 2e, the typical diameter of the fiber was $\sim 650 \text{ nm}$. Thus, the total contact area between the ion gel and the ZTO fibers was calculated to be $\sim 6792.5 \mu\text{m}^2$.

To clarify this point, we added one sentence in the revised manuscript and one figure in the revised Supplementary Information. The reviewer may refer to the following content in the revised manuscript and the Supplementary Information:

In the revised manuscript:

Page 7 and line 4: “A ZTO AS used ~ 19 ZTO fibers as channels, and the total contact area between ZTO fibers and ion gel was $\sim 6792.5 \mu\text{m}^2$ (Supplementary Fig. 1).”

In the revised Supplementary Information:

Supplementary Figure 1. Statistics information about number of ZTO fibers and their coverage area of the surface. a Digital image of an artificial synapse. **b, c, d** Optical microscopy images of the device at $5\times$ (**b**), $20\times$ (**c**), and $50\times$ magnification (**d**). **c** corresponds to the area in the dotted box of **b**, and **d** corresponds the dotted box in **c**. Note: The distance between two adjacent interdigital electrodes is $\sim 50 \mu\text{m}$. The fibers between two adjacent interdigital electrodes are exposed and covered with ion gel, and the rest are covered with gold electrodes. In each ZTO fiber, 11 segments are exposed to ion gel, so the total length of one fiber covered with ion gel is $\sim 550 \mu\text{m}$. In addition, ~ 288 Au/ZTO junctions in total engaged in the formation of postsynaptic current.

Comment 6 from Reviewer #3:

Furthermore, in lines 437-446, it was claimed that ZTO fibers were printed on the Si/SiO₂ substrate surface, followed by fabrication of Au electrodes, and then transfer of ion gel. We suppose that SiO₂ is developed on Si. Please clarify and explain the method of deposition of SiO₂ and its possible thickness. Therefore, according to the method section, the structure is “Si/SiO₂-ZTO fiber/Au electrode/Ion gel”. But, if authors return to the graphical image of synaptic device, it shows that Au electrodes are deposited on Si/SiO₂ substrate, followed by development of ZTO fiber arrays on top of Au electrode. Therefore, authors should provide the correct graphical scheme of their ZTO fiber based artificial synaptic system in Figure 1, Figure 2, Figure 3, and Figure 4.

Response: Thank you very much for your valuable comment and kind suggestion. The SiO₂ layer (~ 300 nm thick) of the Si/SiO₂ wafer was obtained by wet thermal oxidation of Si. The 4-inch wafers of Si/SiO₂ (~ 500 μm thick) were purchased from Tebo Technology Co., Ltd., and used without further processing. Thus, the structure of the ZTO artificial synapse is “Si/SiO₂/ZTO fiber/Au electrode/Ion gel”. We feel sorry for the confusion in the graphical image of the synaptic device. We have corrected the graphical scheme of the ZTO artificial synapse in Figure 1, Figure 2, Figure 3, Figure 4, and Figure 5.

To clarify these points, we have added one sentence and corrected the graphical scheme of the ZTO artificial synapse in Figure 1, Figure 2, Figure 3, Figure 4, and Figure 5 in the revised manuscript. The reviewer may refer to the following content in the revised manuscript:

Page 22 and line 4: “The 4-inch Si/SiO₂ wafers (~ 500 μm thick) with a SiO₂ layer of ~ 300 nm were purchased from Tebo Technology Co., Ltd.”

Figures: Figure 1, Figure 2b, Figure 3b, Figure 4c, and Figure 5c.

Comment 7 from Reviewer #3:

Moreover, if the Au electrodes are developed on top of the ZTO fibers, authors should provide a clear information that explain how many Au/ZTO junctions are engaged in the formation of postsynaptic current, and also elaborate the way that pre-synaptic current is applied. Generally, the materials and method section should clearly explain the details of fabrication and measurement process.

Response: We sincerely appreciate your valuable comment. The Au electrodes were developed on top of the ZTO fibers. For each artificial synapse, there are ~ 288 Au/ZTO junctions engaged in the formation of postsynaptic current. This point has been clarified in the Materials and Method section and Supplementary Figure 1. The reviewer may refer to the corresponding content in the revised manuscript and Supplementary Information. We also performed additional experiments to investigate the relationship

between the number of ZTO fibers and aEPSC (Supplementary Fig. 2). The aEPSC increased as the number of ZTO fibers increased.

About the method to apply presynaptic spikes: A metal probe that contacted the ion gel was used as the presynaptic input terminal to apply the presynaptic spikes.

According to the reviewer's suggestion, we have added some experimental details to the Materials and Method section. However, to control the length of the article, other experimental and technical details, such as Fabrication of electrochromic device, Material characterization and device measurements, First-principles calculations, Construction of artificial synapse-amplifier circuit-electrochromic actuator system were placed in the Supplementary Information. We have mentioned in the Methods section that "The Supplementary Information displays additional experimental details".

We have added several sentences in the revised manuscript and one figure in the Supplementary Information. The reviewer may refer to the following content in the revised manuscript and Supplementary Information:

In the revised manuscript:

Page 7 and line 6: "The aEPSC increased as the number of ZTO fibers increased (Supplementary Fig. 2)."

Page 23 and line 5: "A metal probe that contacted the ion gel was used as the presynaptic input terminal to apply the presynaptic spikes. For an artificial synapse, there are ~ 19 fibers and correspondingly ~ 288 Au/ZTO junctions engaged in the formation of postsynaptic current."

In the revised Supplementary Information:

Supplementary Figure 2. a $\Delta aEPSC$ of ZTO-3:7 artificial synapse with 3, 5, 7, or 19 ZTO fibers. b $\Delta aEPSC$ peak of ZTO-3:7 artificial synapse with vs number (3, 5, 7, 19) of ZTO fibers.

Comment 8 from Reviewer #3:

In Figure 2g, authors should provide the SAED pattern or high-resolution fast Fourier-transform (FFT) pattern of ZTO crystalline structure to confirm their points.

Response: We sincerely appreciate your meaningful comment. In response, we have provided the selected area electron diffraction (SAED) pattern of the ZTO crystalline structure as the inset of the high-resolution TEM image (Fig. 2g). In addition, the SAED patterns of ZTO-7:3 and ZTO-1:1 fibers have also been provided (Supplementary Fig. 7).

We have added two sentences and modified the corresponding figures in the revised manuscript and the Supplementary Information. The reviewer may refer to the following content in the revised manuscript and Supplementary Information:

In the revised manuscript:

Page 7 and line 12: “The selected area electron diffraction (SAED) pattern of ZTO-3:7 fibers confirmed their polycrystalline structure (inset of Fig. 2g).”

Page 8 and line 17: “...was confirmed by HRTEM images and SAED patterns (Supplementary Fig. 7).”

Figure 2g:

“Fig.2 ... g HRTEM image of ZTO-3:7 fibers. Inset: SAED pattern of ZTO-3:7 fibers. ...”

In the revised Supplementary Information:

Supplementary Figure 7. High-resolution TEM images. a ZTO-7:3 fibers. **b** ZTO-1:1 fibers. Inset: Selected area electron diffraction (SAED) patterns of the corresponding samples.

Comment 9 from Reviewer #3:

Figure 2I is not readable. Either provide a clear version or transfer the file into the

supplementary information section.

Response: Thank you for your kind suggestions. We apologize for this confusion. In response, we have modified Figure 21 to make it clear. The reviewer may refer to the following content in the revised manuscript:

The new Figure 21:

Comment 10 from Reviewer #3:

Line 132...please edit the sentence. Was deconvoluted.

Response: Thank you for your helpful comment. We have rewritten this sentence in the revised manuscript. Reviewer may refer to the following content in the revised manuscript:

Page 7 and line 21: “The O 1s peak of ZTO-3:7 fibers was divided into three peaks, approximately centered at 530 eV (O_a), 531 eV (O_b), and 532 eV (O_c) (Fig. 2i).”

Comment 11 from Reviewer #3:

What does PEDOT:PPS represent?

Response: Thank you very much for your question. We neglected to give the full name. PEDOT:PSS represents Poly(3,4-ethylenedioxythiophene):poly(styrene sulfonate) which is one of the most widely used organic materials^{R23}. PEDOT:PSS is chemically stable in air and water, has high electrical conductivity, and has superior optical transparency. Thus, it is extensively used as the conductor and electrode in various electronic devices, such as organic light-emitting diodes, solar cells, and electrochromic devices^{R24}.

We have provided the full name of PEDOT:PSS at first mention in the revised manuscript. The reviewer may refer to the following content in the revised manuscript:

Page 7 and line 33: “poly(3,4-ethylenedioxythiophene):poly(styrene sulfonate) (PEDOT:PSS) films ”

Reference

- R23. Sun, K. *et al.* Review on application of PEDOTs and PEDOT:PSS in energy conversion and storage devices. *J. Mater. Sci. Mater. El.* **26**, 4438–4462 (2015).
- R24. Volkov, A. V. *et al.* Understanding the capacitance of PEDOT:PSS. *Adv. Funct. Mater.* **27**, 1700329 (2017).

Comment 12 from Reviewer #3:

How did you calculate the gate capacitance in Note 1?

Response: Thank you for your question. The gate capacitance in Note 1 was characterized using a semiconductor parameter analyzer (B1500A, Agilent). For the test sample, the ion gel was sandwiched between Al and ITO electrodes (inset of Supplementary Fig. 12). At 1 kHz, the frequency-dependent capacitance of the ion gel is $\sim 2.31 \mu\text{F}/\text{cm}^2$, due to the formation of electrical double layers induced by mobile ions^{R25,26}.

To clarify this point, we have added several sentences and one figure in the revised manuscript and the Supplementary Information. The reviewer may refer to the following content in the revised manuscript and the Supplementary Information:

In the revised manuscript:

Page 9 and line 18: "... (Supplementary Note 1, **Supplementary Fig. 12**)"

In the revised Supplementary Information:

Page 2 and line 23: "The frequency-dependent capacitance of the ion gel was characterized using a semiconductor parameter analyzer (B1500A, Agilent)."

Page 4 and line 6: "At 1 kHz, the frequency-dependent capacitance of the ion gel is $\sim 2.31 \mu\text{F}/\text{cm}^2$ (Supplementary Fig. 12), due to the formation of electrical double layers induced by mobile ions^{S7, 8}."

Reference:

- S7. Choi, H. H., Cho, K., Frisbie, C. D., Siringhaus, H. & Podzorov, V. Critical assessment of charge mobility extraction in FETs. *Nat. Mater.* **17**, 2–7 (2017).
- S8. Jiang, C. *et al.* Mammalian-brain-inspired neuromorphic motion-cognition nerve achieves cross-modal perceptual enhancement. *Nat. Commun.* **14**, 1344 (2023).

Supplementary Fig. 12:

Supplementary Figure 12. Frequency-dependent capacitance of the ion gel. The ion gel was sandwiched between Al and ITO electrodes. Inset: Schematic structure of the test sample.

Reference

- R25. Choi, H. H., Cho, K., Frisbie, C. D., Sirringhaus, H. & Podzorov, V. Critical assessment of charge mobility extraction in FETs. *Nat. Mater.* **17**, 2–7 (2017).
- R26. Jiang, C. *et al.* Mammalian-brain-inspired neuromorphic motion-cognition nerve achieves cross-modal perceptual enhancement. *Nat. Commun.* **14**, 1344 (2023).

Comment 13 from Reviewer #3:

It was explained that the structure of ZTO 1:1 fibers was disordered after simulated annealing? It is the first time and the last time that manuscript talk about annealing? Is there any annealing process in fabrication of ZTO fibers? Please clarify this issue.

Response: We sincerely appreciate your comment and apologize for the confusion. It is the first time that the manuscript talks about annealing. Actually, the fabrication of ZTO fibers entails an annealing process, and the simulated annealing temperature is consistent with the annealing process in the fabrication of ZTO fibers. We have mentioned in the section “Methods-ZTO fibers fabrication” in the original manuscript, that “After being printed, the samples were calcinated in a muffle furnace at 500 °C for 2 h.” The calcination for ZTO fibers is an annealing process.

To clarify this point, we have changed the term “calcinated” to “annealed”, and added several sentences in the revised manuscript. The reviewer may refer to the following content in the revised manuscript:

Page 9 and line 28: “The annealing temperature in the simulation was consistent with the annealing temperature used during the fabrication of ZTO fibers. In the simulation, after annealing, the structure of ZTO-1:1 fibers was more disordered than those of ZTO-7:3 and ZTO-3:7 (Supplementary Fig. 13b).”

Page 22 and line 19: “After being printed, the samples were annealed in a muffle

furnace at 500 °C for 2 h.”

Page 22 and line 24: “The printed ZTO fibers were annealed in a muffle furnace at 500 °C for 2 h.”

Comment 14 from Reviewer #3:

It is necessary to practically measure the bandgap of ZTO fibers and compare it with the result of DOS in Figure 2n and Figure S 11.

Response: We sincerely appreciate this valuable comment. We have performed additional experiments to determine the band gap E_g of ZTO fibers. For this purpose, we adopted methods that use optical absorbance data plotted appropriately with respect to energy^{R27,28}. The experimental results demonstrated that the band gaps of ZTO-7:3, 1:1 and 3:7 fibers were 3.62, 3.54 and 3.66 eV, respectively (Supplementary Note 2, Supplementary Figure 16, and Supplementary Table 3). The band gaps of ZTO-7:3, 1:1 and 3:7 fibers obtained from simulation were 0.75, 0.53 and 1.58 eV, which are similar to other calculations^{R29,30}. The simulation values are smaller than the experimental values due to the limitation of DFT^{R29,31}. However, the variation trend of band gaps obtained from experiments is consistent with that from simulation. The band gaps increased as Zn:Sn molar ratio decreased. The band gap of ZTO-1:1 fibers was the smallest because amorphization of the crystal structure induces defects in the system.

In response to this comment, we have added several sentences and references, one Note, one figure, and one table in the revised manuscript and the Supplementary Information. The reviewer may refer to the following content in the revised manuscript and the Supplementary Information:

In the revised manuscript:

Page 10 and line 1: “The density of states (DOS) for the ZTO-7:3 (Supplementary Fig.15a), ZTO-1:1 (Supplementary Fig.15b), and ZTO-3:7 (Fig. 2o) fibers shows that the band gaps of three samples were 0.75, 0.53 and 1.58 eV, respectively, which are similar to other calculations^{41,42}. The simulation values are smaller than the corresponding experimental values of 3.62, 3.54, and 3.66 eV (Supplementary Note 2, Supplementary Fig. 16); this difference can be attributed to the limitation of DFT^{41, 43}. However, the variation trend of band gaps obtained from experiments is consistent with that from the simulation (Supplementary Table 3).”

Reference:

41. Xia, C., Wang, F. & Hu, C. Theoretical and experimental studies on electronic structure and optical properties of Cu-doped ZnO. *J. Alloy. Compd.* **589**, 604–608 (2014).
42. Gou, H., Gao, F. & Zhang, J. Structural identification, electronic and optical properties of ZnSnO₃: First principle calculations. *Comp. Mater. Sci.* **49**, 552–555 (2010).

43. Li, P. *et al.* First-principle studies on the conductive behaviors of Ga, N single-doped and Ga-N codoped ZnO. *Comp. Mater. Sci.* **50**, 153–157 (2010).

In the revised Supplementary Information:

Supplementary Note 2: Calculation of band gap of ZTO fibers

The band gap E_g of ZTO fibers was extracted from the following equation^{S9, 10}:

$$\alpha hv = A(hv - E_g)^n$$

where α is the absorption coefficient, h is Planck’s constant, v is the photon’s frequency, A is a constant, E_g is the band gap, and n is 0.5 here.

Supplementary Figure 16. Band gap measurement of ZTO fibers. Plot of $(Ahv)^2$ as a function of photon energy hv for three samples.

Supplementary Table 3. Band gap of ZTO fibers with different Zn:Sn molar ratios obtained by experiment and simulation

Samples	Bandgap (eV) Experiment	Bandgap (eV) Simulation
ZTO 7:3	3.62	0.75
ZTO 1:1	3.54	0.53
ZTO 3:7	3.66	1.58

Reference:

- S9. Coulter, J. B. & Birnie, D. P. Assessing Tauc plot slope quantification: ZnO thin films as a model system. *Phys. Status Solidi B* **255**, 1700393 (2018).
- S10. Mao, J.-Y. *et al.* Lead-free monocrystalline perovskite resistive switching device for temporal information processing. *Nano Energy* **71**, 104616 (2020).

Reference

- R27. Coulter, J. B. & Birnie, D. P. Assessing Tauc plot slope quantification: ZnO thin films as a model system. *Phys. Status Solidi B* **255**, 1700393 (2018).
- R28. Mao, J.-Y. *et al.* Lead-free monocrystalline perovskite resistive switching device for temporal information processing. *Nano Energy* **71**, 104616 (2020).
- R29. Xia, C., Wang, F. & Hu, C. Theoretical and experimental studies on electronic structure and optical properties of Cu-doped ZnO. *J. Alloy. Compd.* **589**, 604–608 (2014).
- R30. Gou, H., Gao, F. & Zhang, J. Structural identification, electronic and optical properties of ZnSnO₃: First principle calculations. *Comp. Mater. Sci.* **49**, 552–555 (2010).
- R31. Li, P. *et al.* First-principle studies on the conductive behaviors of Ga, N single-doped and Ga–N codoped ZnO. *Comp. Mater. Sci.* **50**, 153–157 (2010).

Comment 15 from Reviewer #3:

Please elaborate how the mechanical stimuli turn on the vibration sensors, including the frequency of vibration and its information.

Response: We sincerely appreciate your insightful comment. In response to this comment, we have provided enough details in the revised manuscript and Supplementary Information. The vibration sensor-oscillation circuit acts as a receptor to convert mechanical stimuli to electrical pulses. When a finger or a foreign object touches the vibration sensor, it acts as a normally closed device that conducts at this time, making the power-supply side of the oscillator circuit turn on and output a high-frequency pulse (Figure R4). Figure R5 shows the electrical performance of the vibration sensor-oscillator circuit.

Figure R4. Schematic diagram of operation and working mechanism of vibration

sensor.

Figure R5. Characterization of the electrical performance of the vibration sensor-oscillator circuit.

We added some details in the revised manuscript and Supplementary Information. The reviewer may refer to the following content in the revised manuscript and the Supplementary Information:

In the revised manuscript:

Page 17 and line 20: “The statistical curve of signal output over time for each part of the artificial corneal reflex arc under mechanical stimulation was determined (Supplementary Fig. 24). When a finger or a foreign object touches the vibration sensor, it acts as a normally closed device that conducts at this time, making the power-supply side of the oscillator circuit turn on and output a high-frequency spikes pulse. The output spike signals are used as the presynaptic stimulus to trigger the postsynaptic current of the artificial synapse, then an amplifier circuit outputs a voltage signal, which is used as the modulation signal of the electrochromic unit. Gradual changes of electrochromic actuators under the real-time touch can be seen in Supplementary Movie 1.”

In the revised Supplementary Information: Supplementary Figures 21 and 24.

Supplementary Figure 21. Sensors and sensor-oscillation circuits. **a** Vibration sensor and corresponding sensor-oscillation circuit. **b** Schematic of operation and working mechanism of vibration sensor. **c** Light sensor and corresponding sensor-oscillation circuit. **d** Schematic of operation and working mechanism of light sensor. The sensor-oscillation circuit can encode stimulus information by spike frequency. The somatosensory and light neural communication by integrating ZTO artificial synapses with sensors (vibration-threshold or light-dependent resistor) by an improving sensor-oscillation circuit. A commercially-available vibration sensor (SW-180 10P) and photosensitive resistor (T5506) were used to ensure the operational stability of the sensor-oscillation circuit.

Supplementary Figure 24. Statistical curve of signal output over time for each part of

the artificial corneal reflex arc under mechanical stimulation.

Comment 16 from Reviewer #3:

In Figure 5f, a possible application of the bio-compatible artificial cornea is explained, however, there is no evidence or outcome about this function. Therefore, there is no point to allocate a figure to a concept which is not developed.

Response: Thank you very much for your valuable comment. We agree with the reviewer's comment and have removed the original Figure 5f.

Comment 17 from Reviewer #3:

I strongly recommend to add video files to shows the performance of artificial cornea with feelings. You may provide several brief video files that shows the gradual changers of electro-chromic actuators at different intensity of optical inputs.

Response: We sincerely appreciate your suggestion. We have added three brief video files to show the performance of the artificially-intelligent cornea. The reviewer may refer to the following content in the revised manuscript and the Supplementary Movies:

In the revised manuscript:

Page 17 and line 28: "Gradual changes of electrochromic actuators under the real-time touch can be seen in Supplementary Movie 1."

Page 18 and line 5: "(Fig. 4i, Supplementary Fig. 25, Supplementary Movie 2)"

Page 20 and line 20: "(Fig. 4f, Supplementary Movie 3)"

In the Supplementary Movie:

Supplementary Movie 1: Gradual changes of electrochromic actuators under the real-time touch;

Supplementary Movie 2: States of electrochromic actuators under bilateral reflex, ipsilateral reflex and contralateral reflex;

Supplementary Movie 3: Gradual changes of electrochromic actuators under different light input intensities.

REVIEWER COMMENTS

Reviewer #1 (Remarks to the Author):

This revised version of the manuscript incorporates many suggestions of the reviewers and has improved significantly.

Reviewer #2 (Remarks to the Author):

The revision has mostly addressed my comments, making the paper more suitable for acceptance. But there are still two things that the authors should revise:

1. Regarding the response speed of the electrochromic device in reference two the speed of eyelid closure, I don't think it is reasonable to use the shortest pulse duration that can be responded by the device to make an analogy to the eyelid closure time. The eyelid closure makes complete transition from fully open to fully close. However, one short pulse only changes the light transmission of the electrochromic device a little bit. As such, the "4.3 s" that corresponds to the 90% transition of the color change is a more suitable parameter to use in this context. Therefore, I think the following statement made by the authors is misleading and should be corrected: "...and it can respond to the voltage with a duration of ~ 50 ms (Supplementary Fig. 23), which is less than the duration of eyelid closure (100 ~ 150 ms)⁶⁴. Thus, the response speed of the device can replicate the eyelid-closure process."

2. For the added citations of the recent stretchable neuromorphic device works, the authors only included the papers that are along the direction of "neuromorphic sensing" applications, but didn't include the works in the direction of neuromorphic computing for AI and ML, especially for processing health data. Since the main stream application of neuromorphic computing is to implement AI, representative works on stretchable neuromorphic devices for on-body data processing with artificial intelligence should also be cited, so that this discussion is complete.

Reviewer #3 (Remarks to the Author):

The authors of the manuscript entitled "An artificially-intelligent cornea with tactile sensation enables sensory expansion and interaction" have replied my comment sufficiently. The present style of manuscript can be considered for further processing towards acceptance.

Response to Reviewers' Comments

(Manuscript ID: NCOMMS-23-17850A)

We would like to sincerely thank the editor and reviewers once again for considering and reviewing our manuscript. Your valuable comments and suggestions are helpful in improving the quality of our work.

The comments of the reviewers are in black, our responses are **in blue**, and the revisions to the manuscript and Supplementary Information are highlighted **in red**.

Reviewer #1:

This revised version of the manuscript incorporates many suggestions of the reviewers and has improved significantly.

Response: The positive comment on our work from the reviewer is highly appreciated. Your valuable suggestions and insightful comments helped us to significantly improve the quality of this work.

Reviewer #2:

The revision has mostly addressed my comments, making the paper more suitable for acceptance. But there are still two things that the authors should revise:

Response: Thank you very much for your meaningful comments and suggestions that helped us to improve the quality of this work. In response to the reviewer's new comments, we have revised the corresponding contents. The point-by-point responses to the reviewer's comments are presented below.

Comment 1 from Reviewer #2:

Regarding the response speed of the electrochromic device in reference two the speed of eyelid closure, I don't think it is reasonable to use the shortest pulse duration that can be responded by the device to make an analogy to the eyelid closure time. The eyelid closure makes complete transition from fully open to fully close. However, one short pulse only changes the light transmission of the electrochromic device a little bit. As such, the "4.3 s" that corresponds to the 90% transition of the color change is a more suitable parameter to use in this context. Therefore, I think the following statement made by the authors is misleading and should be corrected: "...and it can respond to the voltage with a duration of ~ 50 ms (Supplementary Fig. 23), which is less than the duration of eyelid closure (100 ~ 150 ms)⁶⁴. Thus, the response speed of the device can

replicate the eyelid-closure process."

Response: We sincerely appreciate this valuable comment and agree with the reviewer's point. In response to this comment, we have revised the corresponding content. The reviewer may refer to the following content in the revised manuscript:

Page 17 and line 13: "The response time of the electrochromic device, defined as the period to achieve a 90 % change in transmittance, is ~ 4.3 s under 3 V (Supplementary Fig. 23a, b). Considering the duration of eyelid closure is approximately 100 ~ 150 ms⁶⁷, the change in transmittance of the electrochromic device in 150 ms was calculated as ~ 9 % (Supplementary Fig. 23c). Thus, the response speed of the device can partially replicate the eyelid-closure process."

Comment 2 from Reviewer #2:

For the added citations of the recent stretchable neuromorphic device works, the authors only included the papers that are along the direction of "neuromorphic sensing" applications, but didn't include the works in the direction of neuromorphic computing for AI and ML, especially for processing health data. Since the main stream application of neuromorphic computing is to implement AI, representative works on stretchable neuromorphic devices for on-body data processing with artificial intelligence should also be cited, so that this discussion is complete.

Response: Thank you very much for your meaningful suggestions. In response to this comment, we have revised the corresponding content and added three references in the revised manuscript. The reviewer may refer to the following content in the revised manuscript:

Page 14 and line 13: "The flexible and stretchable properties of artificial synapses are of great significance for wearable and human-interface applications⁵³⁻⁵⁶, **neuromorphic computing for artificial intelligence^{57,58}, especially on-body data processing⁵⁹.**"

Reference:

57. Wang, Y., Cao, M., Bian, J., Li, Q. & Su, J. Flexible ZnO nanosheet-based artificial synapses prepared by low-temperature process for high recognition accuracy neuromorphic computing. *Adv. Funct. Mater.* **32**, 2209907 (2022).
58. Huang, J. *et al.* Flexible, transparent, and wafer - scale artificial synapse array based on TiO_x/Ti₃C₂T_x film for neuromorphic computing. *Adv. Mater.* **35**, 2303737 (2023).
59. Dai, S. *et al.* Intrinsically stretchable neuromorphic devices for on-body processing of health data with artificial intelligence. *Matter* **5**, 3375-3390 (2022).

Reviewer #3:

The authors of the manuscript entitled "An artificially-intelligent cornea with tactile sensation enables sensory expansion and interaction" have replied my comment sufficiently. The present style of manuscript can be considered for further processing towards acceptance.

Response: Thank you very much for the time and efforts dedicated to our manuscript. We sincerely appreciate your valuable comments and constructive suggestions that helped us significantly improve this work.

REVIEWERS' COMMENTS

Reviewer #2 (Remarks to the Author):

This paper has now addressed all the comments, and can be recommended for publication.

Response to Reviewers' Comments

(Manuscript ID: NCOMMS-23-17850B)

We would like to sincerely thank the editor and reviewers for considering and reviewing our manuscript. Your insightful comments and suggestions are helpful in improving the quality of our work.

The comments of the reviewer are in black, and our responses are in blue.

Reviewer #2:

This paper has now addressed all the comments, and can be recommended for publication.

Response: Thank you very much for recommending the publication of our manuscript. Your valuable comments helped us to significantly improve the quality of this work.